# T4 DNA polymerase prevents deleterious on-target DNA damage and enhances precise CRISPR editing

Qiaoyan Yang [1], Jonathan S Abebe[1], Michelle Mai[1], Gabriella Rudy [1], Sang Y Kim[2], Orrin Devinsky [3] & Chengzu Long [1]✉

## Abstract

Unintended on-target chromosomal alterations induced by CRISPR/Cas9 in mammalian cells are common, particularly large deletions and chromosomal translocations, and present a safety challenge for genome editing. Thus, there is still an unmet need to develop safer and more efficient editing tools. We screened diverse DNA polymerases of distinct origins and identified a T4 DNA polymerase derived from phage T4 that strongly prevents undesired on-target damage while increasing the proportion of precise 1- to 2-base-pair insertions generated during CRISPR/Cas9 editing (termed CasPlus). CasPlus induced substantially fewer on-target large deletions while increasing the efficiency of correcting common frameshift mutations in DMD and restored higher level of dystrophin expression than Cas9-alone in human cardiomyocytes. Moreover, CasPlus greatly reduced the frequency of on-target large deletions during mouse germline editing. In multiplexed guide RNAs mediating gene editing, CasPlus repressed chromosomal translocations while maintaining gene disruption efficiency that was higher or comparable to Cas9 in primary human T cells. Therefore, CasPlus offers a safer and more efficient gene editing strategy to treat pathogenic variants or to introduce genetic modifications in human applications.

**Keywords** DNA Polymerase; CRISPR/Cas9; On-target Damage; DMD; T Cell
**Subject Category** Biotechnology & Synthetic Biology

## Introduction

The engineered clustered regularly interspaced short palindromic repeats/CRISPR-associated protein 9 (CRISPR/Cas9)-mediated genome editing has revolutionized genetics (Cong et al, 2013; Jinek et al, 2012; Jinek et al, 2013; Mali et al, 2013). However, Cas9 can generate undesired on-target large deletions (Kosicki et al, 2018; Nahmad et al, 2022b), chromosomal translocations

(Stadtmauer et al, 2020), chromothripsis (Leibowitz et al, 2021), and other complex chromosomal rearrangements (Boutin et al, 2022), as well as off-target effect. Although numerous strategies have been developed to minimize CRISPR/Cas9-mediated off-target effects (Uddin et al, 2020), few approaches can mitigate collateral on-target DNA damage (Bothmer et al, 2020; Xin et al, 2022; Yin et al, 2022; Yoo et al, 2022). Base editing and prime editing are CRISPR/Cas-based gene editing technologies which can efficiently and precisely edit genes, but both have limitations. Base editing is not designed to mediate indels and has off-target effects on DNA and RNA. Prime editing has lower editing efficiency than Cas9 when targeting identical genome sites (Anzalone et al, 2020; Anzalone et al, 2019; Gaudelli et al, 2017; Kim et al, 2021; Komor et al, 2016). Prime editing efficiency and fidelity are inhibited by DNA mismatch repair (MMR), which is a highly conserved DNA repair pathway and maintains genomic stability (Ferreira da Silva et al, 2022). Clinical trials with CRISPR/Cas9 therapies are underway (2024), but new tools to efficiently introduce precise edits with minimal byproducts and on-target damage would further advance the genome editing field.

Cas9 cleaves target DNA producing double-strand breaks (DSBs) with blunt ends or staggered ends with 5′ overhangs (Shi et al, 2019). Repairing these ends without exogenous templates usually occurs via non-homologous end joining (cNHEJ) or microhomology-mediated end joining (MMEJ) (Chang et al, 2017). The specific repair pathway determines Cas9 editing outcomes. MMEJ repair requires an end resection and micro-homology sequences, and often results in deletions. MMEJ is associated with Cas9-induced on-target large deletions, chromosome translocations and rearrangement (Kosicki et al, 2022; Owens et al, 2019; Sfeir and Symington, 2015). cNHEJ repair directly joins two compatible ends without or with minor end resection, leading to small insertions or deletions (indels) (Mao et al, 2008; Sfeir and Symington, 2015). Comprehensive analyses on Cas9 on-target edits reveal small insertions produced by exogenous template-free Cas9 editing are precise and predictable (Allen et al, 2018; Leenay et al, 2019; Shen et al, 2018). The frequency and pattern of Cas9-generated small insertions depend on the local sequences surrounding the Cas9 cut site (templated insertions) (Shi et al, 2019). Current tools cannot fully control the outcomes of 1- to 2-base-pair (bp) insertions, which can be used to reframe or knockout

[1]NYU Cardiovascular Research Center, Leon H. Charney Division of Cardiology, Department of Medicine, NYU Langone Health, New York, NY, USA. [2]Department of Pathology, NYU Langone Health, New York, NY, USA. [3]New York University Langone Comprehensive Epilepsy Center, NYU Langone Health, New York, NY, USA.
✉E-mail: Chengzu.Long@nyulangone.org

genes (Bermudez-Cabrera et al, 2021). In budding yeast, Cas9-induced 1–3-bp insertions are attributed to DNA polymerase 4 (Pol 4) (Lemos et al, 2018). Thus, systematic screening gap-filling DNA polymerases may identify enzymes that favor cNHEJ-mediated small insertions during Cas9 editing in mammalian cells. The competition between fill-in and end resection processes in DSB with staggered ends (Cejka and Symington, 2021; Kosicki et al, 2022) suggests that gap-filling DNA polymerases may also inhibit MMEJ-mediated on-target large deletions and chromosome translocations.

Duchenne muscular dystrophy (DMD) is characterized by the degeneration of cardiac and skeletal muscles (O'Brien and Kunkel, 2001) and results from mutations in the X-linked dystrophin gene (*DMD*) (Muntoni et al, 2003). Deletion of one or more exons is the most common mutation type (Duan et al, 2021). Dilated cardiomyopathy (DCM) is a common and lethal feature (Adorisio et al, 2020). We used CRISPR/Cas9 to correct or bypass the *DMD* mutations in cultured human cells and *mdx* mice (Amoasii et al, 2017; Long et al, 2016; Long et al, 2018; Long et al, 2014; Olson, 2021; Zhang et al, 2020; Zhang et al, 2022). However, complex rearrangements can arise during *DMD* editing due to the repetitive elements, stem-loop structures, and preexisting mutations in this unstable genomic region (Nelson et al, 2019; Oshima et al, 2009). Undesired on-target DNA damage at edited *DMD* sites, a safety concern in human therapy, were not addressed. CRISPR/Cas9-engineered Chimeric antigen receptor (CAR) T-cell therapy may overcome conventional CAR-T therapy problems, like Graft-versus-host disease (GvHD), CAR-T-cell exhaustion and limited patient T-cell numbers (Depil et al, 2020; Khan and Sarkar, 2022; Labanieh and Mackall, 2023; Sheridan, 2022a). However, unintended chromosomal abnormalities, such as translocations, pose a safety risk with multiplex gene-edited patient T cells (Bothmer et al, 2020; Poirot et al, 2015; Stadtmauer et al, 2020).

Here, we identified a phage DNA polymerase that markedly reduces the on-target large deletions and chromosomal translocations associated with regular Cas9 editing while maintaining an equal or higher editing efficiency of desired products. Our results demonstrate that our Cas9 and DNA Polymerase, termed CasPlus, can more safely and efficiently correct *DMD* mutation in cardiomyocytes and repress chromosomal translocations in human T cells.

# Results

## T4 and RB69 DNA polymerases favor small insertions over deletions during Cas9 editing

First, we aimed to identify DNA polymerases that can enhance the fill-in step, leading to a ligation process favoring small insertions over deletions during Cas9 editing in mammalian cells (Fig. 1A). We established a stable HEK293T reporter cell line expressing a mutant tdTomato gene with a 1-bp deletion of adenine (A) at position 151 (tdTomato-del151A) (Fig. 1B). Next, we investigated the effects of DNA polymerases on Cas9 editing. We constructed MCP (MS2 bacteriophage coat protein)-tagged vectors expressing human codon-optimized DNA polymerases which can fill in 5' overhang, including (1) X-family polymerases Pol λ, Pol μ, and Pol β of human origin, (2) Pol 4 from yeast (Lemos et al, 2018;

Ramsden and Asagoshi, 2012), (3) family A polymerases Pol I and Klenow fragment (KF) (Bebenek et al, 1990) from bacteria, and (4) B-family polymerases T4 DNA pol from bacteriophage T4 (Liu et al, 2015) (Fig. 1C).

Initially, we delivered MCP-tagged DNA polymerase in trans along with Cas9 and regular guide RNA (System A). We developed a dual fluorescent assay. tdTomato-d151A reporter cells were transfected with a vector containing Cas9/blue fluorescent protein (BFP)/tdTomato-targeted single guide RNA (tdTomato-sgRNA) alone or combined with a second vector expressing DNA polymerases/green fluorescent protein (GFP). We analyzed the reporter gene editing using fluorescence-activated cell sorting (FACS). To ensure an unbiased evaluation of the indel profile alterations, we assessed the indel profiles in all cells expressing both Cas9 and DNA polymerases (BFP$^+$/GFP$^+$) (Fig. 1D). Sequencing of BFP$^+$/GFP$^+$ populations demonstrated that the co-expression of T4 polymerase with Cas9 resulted in ~38% 2-bp insertions, whereas co-expression of Pol I or KF and Cas9 led to ~6–11% 2-bp insertions (Fig. 1E). No indel profile change was observed in other DNA polymerases with Cas9 comparing Cas9-only control (CTR). Similar results were obtained in BFP$^+$/GFP$^+$ populations when RB69 DNA polymerase (Hogg et al, 2006), a B-family DNA polymerase with 61% amino acid homology to T4 DNA polymerase, was co-expressed, but not when the A-family T7 DNA polymerase was used (Hori et al, 1979) (Fig. 1F). The ratio of templated 2-bp insertions to all the 2-bp insertions is up to 89% in T4 or RB69 treated cells (Fig. 1G), suggesting that T4 and RB69 DNA polymerase enables the fill-in of staggered ends created by Cas9 at tdTomato-d151A site.

To reduce the complexity of samples prior to sequencing analysis, we further sorted transfected BFP$^+$GFP$^+$ cells into two subpopulations: one expressing BFP, GFP, and tdTomato (BFP$^+$GFP$^+$tdTomato$^+$) and the other expressing BFP and GFP but not tdTomato (BFP$^+$GFP$^+$tdTomato$^-$) (Appendix Fig. S1A,1B). We utilized targeted deep sequencing to assess the editing profiles in each population. The ratio of BFP$^+$GFP$^+$tdTomato$^+$ to BFP$^+$GFP$^+$ was comparable between cells transfected with the Cas9-only control (CTR) and those co-transfected with Cas9 and distinct DNA polymerases (Appendix Table S1). Sequencing of targeted region revealed that majority of edits observed in BFP$^+$GFP$^+$tdTomato$^+$ populations were 1-bp insertions (Appendix Fig. S1C). Very few $3n + 2$ indels were also detected in BFP$^+$GFP$^+$tdTomato$^+$ populations, indicating that the tdTomato-d151A reporter cell line might contain multiple copies of reporter gene. Targeted sequencing for the BFP$^+$GFP$^+$tdTomato$^-$ populations showed co-expression of T4 DNA polymerase with Cas9 substantially increased the frequency of 2-bp insertions, primarily at the expense of deletions whereas no other DNA polymerase tested obviously altered indel profiles compared to CTR (Appendix Fig. S1C). Similar results were obtained in BFP$^+$GFP$^+$tdTomato$^-$ populations when RB69 DNA polymerase but not in T7 DNA polymerase was co-expressed. (Appendix Fig. S1D). We observed same trends of indel profile alterations in GFP$^+$tdTomato$^-$ populations sorted from tdTomato-d151A reporter cell line, which were transfected either with a vector containing Cas9/GFP/tdTomato-sgRNA alone or in combination with a second vector containing DNA polymerases/GFP (Appendix Fig. S1E–1H). Western blot analysis confirmed expression of DNA polymerases in HEK293T cells (Appendix Fig. S1I). A previous

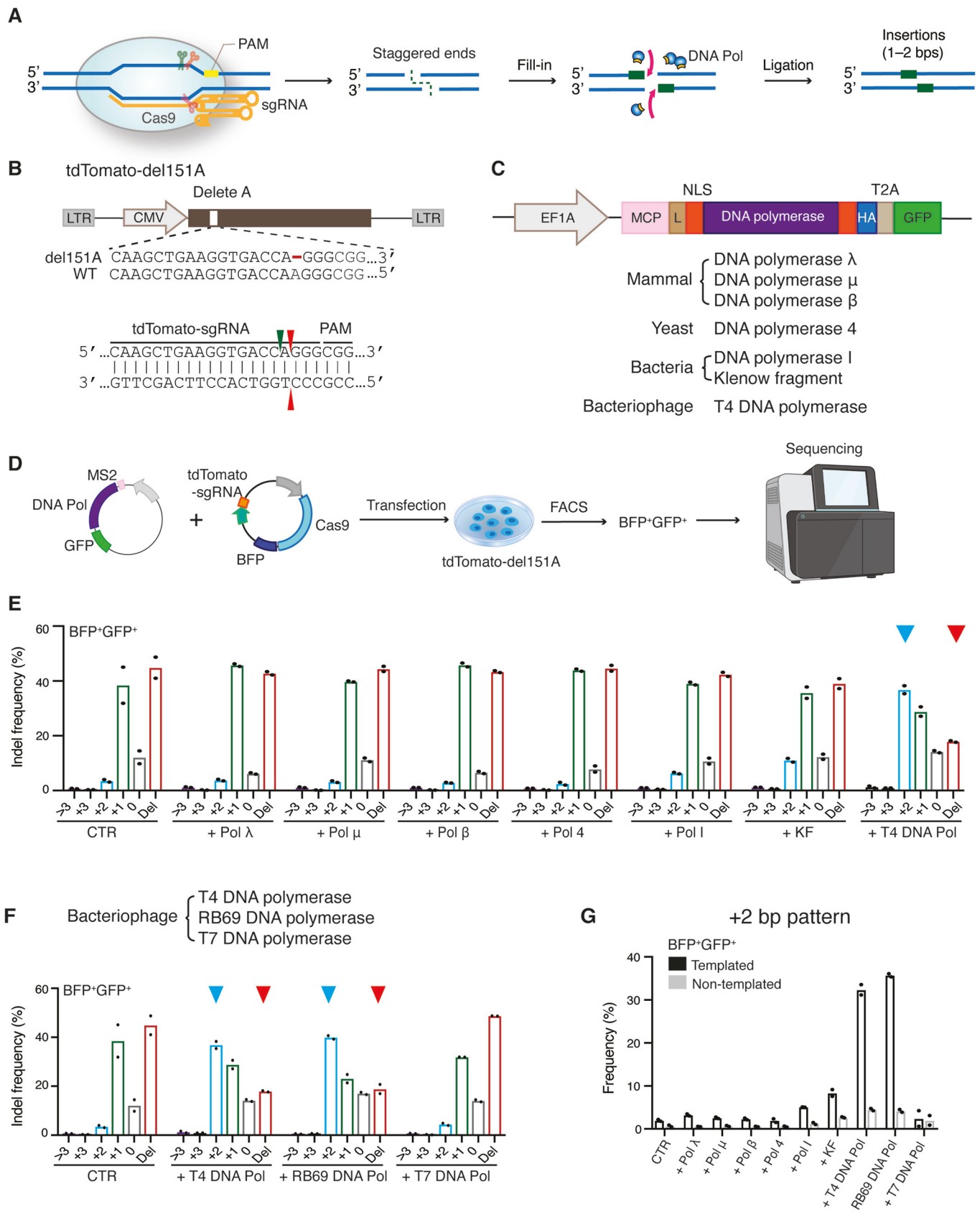

◄ **Figure 1.  T4 and RB69 DNA polymerases favor small insertions over deletions during Cas9 editing.**

(A) Schematic showing the filling-in of Cas9-induced staggered ends by exogenous DNA polymerase. (B) Schematic of the lentiviral vector containing tdTomato-d151A (top) and the gRNA used to target it (bottom). Arrowheads, cut sites; red dash, missing nucleotide; LTR, long terminal repeat. (C) Architecture of vectors expressing distinct DNA polymerases (see "Methods"). L linker, NLS nuclear localization signal. (D) Workflow of the DNA polymerase selection process in tdTomato-d151A reporter cells. tdTomato-d151A reporter cells were transfected with a vector expressing Cas9/BFP/tdTomato-sgRNA and a second vector expressing DNA polymerase/GFP. Three days post-transfection, BFP⁺GFP⁺ cells were sorted using FACS. DNA was subsequently extracted, and the targeted region was amplified for deep sequencing. (E, F) Frequencies of Cas9-induced indels in BFP⁺GFP⁺ populations with or without co-expression of a distinct DNA polymerase. Each dot represents one biological replicate. (G) Patterns of 2-bp insertions in (E, F). Each dot represents one biological replicate. Source data are available online for this figure.

report showed that Pol I and KF combined with Cas9 can increase the frequencies of 1-bp deletions and decrease >1 bp deletions (Yoo et al, 2022). We then further analyzed the proportion of 1-bp deletions to all deletions. At tdTomato-d151A site, we did not detect obvious alterations in 1-bp deletions in Pol I- and KF-treated cells compared to those in CTR (Appendix Fig. S1J), indicating this effect may be guide RNA context-dependent. Subsequent experiments focused on T4 DNA polymerase because it was among the top two most efficient enzymes that favor small insertions in the tested target sites.

We next assessed whether tethering of T4 DNA polymerase to Cas9-induced DSBs using MS2-modifed guide RNA (System B) could further improve small insertions. We engineered tdTomato MS2-sgRNA by incorporating MS2 stem-loops into the classical sgRNA scaffold region (Konermann et al, 2015). We compared the editing profiles at the tdTomato-d151A site produced by co-expression of MCP-tagged T4 DNA polymerase in trans along with Cas9/tdTomato-sgRNA to that with Cas9/tdTomato MS2-sgRNA. We confirmed that the untethered (System A) and MS2-tethered T4 DNA polymerase (System B) was comparable in favoring 2-bp insertions at the tdTomato-d151A site (Appendix Fig. S1K), potentially because of the saturated expression of T4 DNA polymerase when delivered as vectors in vitro, especially after cell sorting. These results indicate MCP-tag is dispensable for T4 DNA polymerase function when delivered in trans with Cas9 in vitro. Therefore, all subsequent in vitro experiments, unless otherwise noted, utilized the untethered system, consisting of MCP-tagged T4 DNA polymerase delivered in trans with Cas9 and guide RNAs in a classical scaffold.

Our next investigation focused on the impact of T4 DNA polymerase on Cas9 editing when fused to the N-terminus or C-terminus of Cas9 via a flexible linker (System C). Unlike co-expression of Cas9 and T4 DNA polymerase in System A and B, Cas9 fused T4 DNA polymerase proteins (T4-L-SpCas9 and Cas9-L-T4) strongly induced deletions at the protospacer-adjacent motif (PAM) distal region at tdTomato-d151A site (Appendix Fig. S1L–1N). An obvious increase of deletions at PAM distal region were also detected at CLCN5 site in T4-L-SpCas9 or Cas9-L-T4 edited cells (Appendix Fig. S1O,1P), which differs from previous observation that Cas9 and DNA Pol I/KF fusion protein can enhance 1-bp deletions at the expense of >1 bp deletions at CLCN5 site with the same guide RNA tested (Yoo et al, 2022). T4 DNA polymerase possesses a polymerase activity that extends the polynucleotide chain, and a 3′-to-5′ proofreading exonuclease activity that excises erroneously incorporated nucleotides (Capson et al, 1992). The 3′-to-5′ exonuclease activity of T4 DNA polymerase is much higher than that of DNA polymerase I or KF (Kucera and Nichols, 2008). Structure studies on the Cas9/guide

RNA and targeted DNA complex reveal that the non-complementary 3′ single-strand DNA (ssDNA) are displaced upon formation of RNA-DNA R-loop (Jore et al, 2011). Hence, we hypothesized that, when T4 DNA polymerase fused to Cas9, its robust exonuclease domain might degrade the displaced 3′ ssDNA at the PAM distal region, leading to relatively large deletions at PAM distal (Appendix Fig. S1Q). Hence, we deactivated the exonuclease domain of T4 DNA polymerase in T4-L-SpCas9 and Cas9-L-T4 by changing Asp-219 reside to Ala (T4-D219A) or inactivated the T4 DNA polymerase domain in T4-L-SpCas9 and Cas9-L-T4 by substituting the Asn-214 reside to Ser (T4-D214S) (Abdus Sattar et al, 1996). High-throughput sequencing (HTS) results demonstrated that the mutation of T4-D219A, but not T4-D214S, eliminated the increased frequency of deletions at the PAM distal region at tdTomato-d151A site induced by T4-L-SpCas9 and Cas9-L-T4 relative to Cas9-only (Appendix Fig. S1R,1S). T4-D219A mutant also abolished the improved deletions induced by T4-L-SpCas9 or SpCas9-L-T4 at CLCN5 site (Appendix Fig. S1T), further confirming this hypothesis.

We next assessed the impact of T4 DNA polymerase on Cas9 editing on the endogenous loci. We designed 26 guide RNAs (gRNAs) targeting multiple pathogenic genes with different chromosomal locations (TS1–TS26; Appendix Fig. S1U). Compared to the Cas9-only treatment control (CTR), T4 DNA polymerase selectively increased 1–2-bp insertions over deletions (≥1-bp) or increased 1–2-bp deletions over ≥3-bp deletions, depending on the gRNA context (Appendix Fig. S1V–1Z); Among the 16 guide RNAs that favored 1-bp insertions with Cas9 and T4 DNA polymerase, 11 (68.8%) contained an adenine (A) or thymine (T) while 5 (31.2%) contained an guanine (G) or cytosine (C) at position −4 (counting the NGG PAM sequences as nucleotides 0–2). Among the ten guide RNAs that biased for 1-bp deletions, six (60%) had an identical nucleotide at positions −4 and −3 while 4 (40%) had a repeating nucleotide at positions −5 and −4. All four guide RNAs that contained a repeating nucleotide at positions −4 and −2 or −5 and −3 led to deletions of the repeating nucleotide and intervening nucleotide (Appendix Fig. S1Z). Collectively, these results reveal that T4 DNA polymerase favors small insertions over deletions, or small deletions (1–2-bp) over ≥3-bp deletions depending on the cleavage site sequences. We refer to this novel gene editing platform—phage T4 DNA polymerase associated with Cas9—as CasPlus editing.

## Improved CasPlus editing efficiency via engineered T4 DNA polymerase

We hypothesized that site-directed mutagenesis on the exonuclease or T4 DNA polymerase domains might affect the fill-in step and

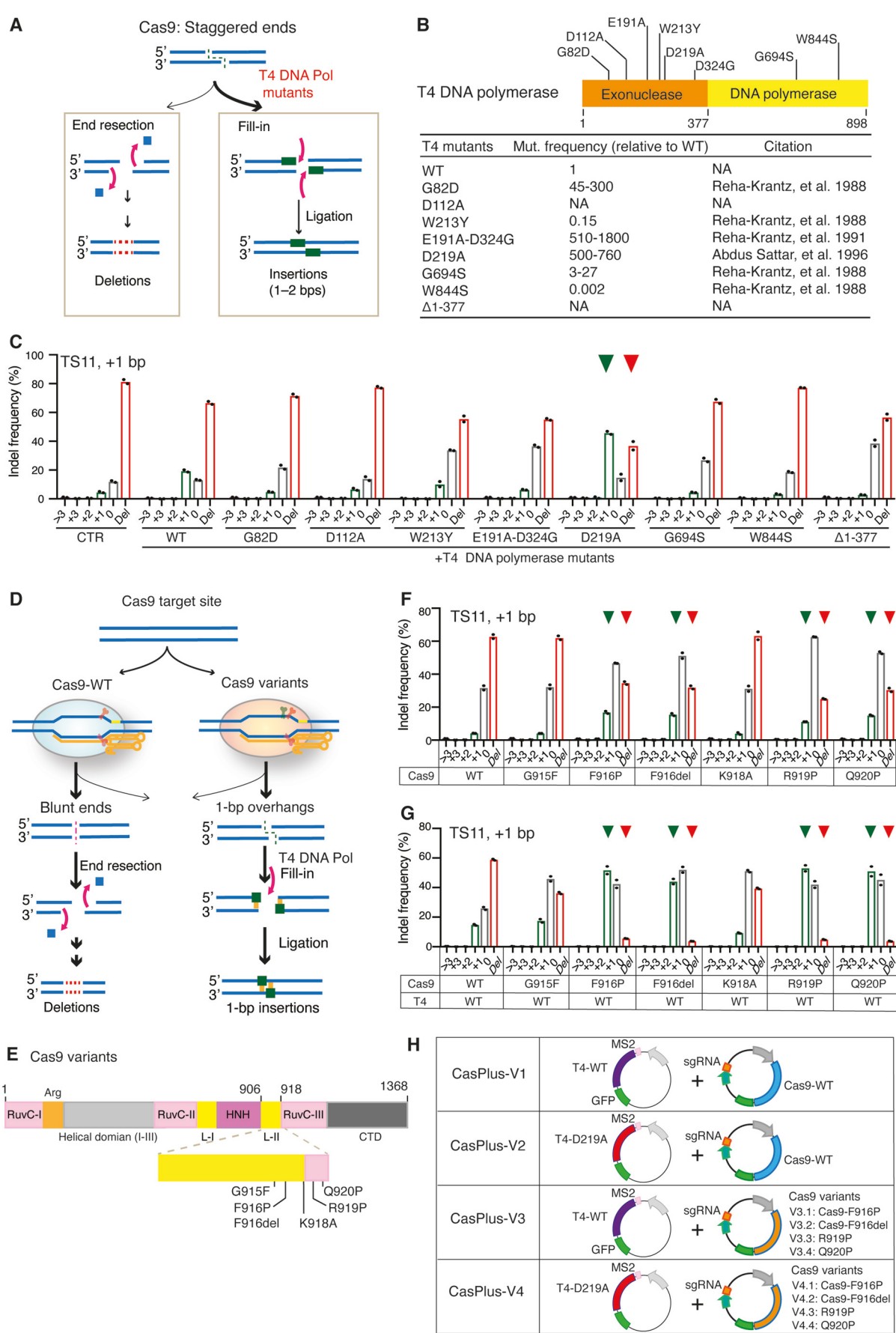

**Figure 2.  Engineered T4 DNA polymerase mutants and Cas9 variants improve CasPlus editing efficiency.**

(A) Cas9-induced staggered ends can be processed for either end resection that generates deletions, or fill-in that produces insertions. T4 DNA polymerase mutants can enhance the fill-in process, thereby increasing 1–2-bp insertions. (B) Schematic showing the amino acid alterations introduced into T4 DNA polymerase; the corresponding relative mutation frequencies are below. NA non-available. (C) Frequency of Cas9-induced indels at TS11 with or without T4 DNA polymerase mutants. Each dot represents one biological replicate. (D) At target sites where Cas9-WT often generates blunt ends, leading to deletions (left), engineered Cas9 variants can preferentially generate staggered ends with 1-bp overhangs, enabling T4 DNA polymerase to produce 1-bp insertions (right). (E) Amino acid alterations introduced within Cas9. (F, G) Frequency of Cas9-induced indels at TS11 in cells transfected with Cas9-WT or distinct Cas9 variants individually (F) or along with T4-WT (G). Each dot represents one biological replicate. (H) Four distinct versions of the T4 DNA polymerase-mediated CasPlus system. Source data are available online for this figure.

change the CasPlus editing efficiency (Fig. 2A). We constructed seven T4 DNA polymerase mutants with a decreased or increased DNA mutation frequency relative to wild-type T4 DNA polymerase (T4-WT), and one N-terminal truncation mutant lacking the exonuclease domain (Δ1–377) (Abdus Sattar et al, 1996; Reha-Krantz, 1988; Reha-Krantz, 1998; Reha-Krantz et al, 1991) (Fig. 2B). Compared to T4-WT, Asp-219 residue to Ala mutant (T4-D219A), had 2.4-fold more 1-bp insertions, at the expense of deletions at TS11 (Fig. 2C). T4-WT and the T4 DNA polymerase mutants exhibited similar 1-bp insertion patterns at TS11 (Appendix Fig. S2A). T4-D219A led to up to sixfold increases in 1-bp insertions at the expense of deletions across 17 genomic sites tested (Appendix Fig. S2B–2D). Mutations changing the Asp-222 residue to Ala in RB69 DNA polymerase (RB69-D222A) (Hogg et al, 2006) also resulted in a twofold to fourfold higher frequency of 1-bp insertions at the TS2, TS11, and TS12 than RB69-WT (Appendix Fig. S2E). Together, these results support that CasPlus editing enhances precise 1-bp insertions, with T4-D219A outperforming T4-WT and RB69-D222A outperforming RB69-WT.

We compared CasPlus and prime editing (PE) in generation of 1-bp insertions at position −4 of TS5 and TS10 (Appendix Fig. S2F). For each target site, we screened four prime editing guide RNAs (pegRNAs) varying in sizes of reverse transcriptase templates (RTT) and prime binding sequences (PBS) and selected the most efficient pegRNA for comparison (Appendix Table S2). PE3 resulted in ~16.1% 1-bp insertions without undesired indels at TS5 and ~8.2% 1-bp insertions without undesired indels at TS10 while Cas9 combined with T4-D219A induced ~72.3% 1-bp insertions and ~4.8% undesired indels at TS5 and ~48.8% 1-bp insertions and ~4.4% undesired indels at TS10 (Appendix Fig. S2G). These results indicated that, at certain defined genome sites, CasPlus editing could be a more efficient approach in introducing precise 1-bp insertions than prime editing.

To further investigate the capabilities of T4 DNA polymerase in favoring small insertions with other Cas proteins, we tested Cas12a, an RNA-guided endonuclease that creates 5′ staggered ends with 5–8-bp overhangs(Zetsche et al, 2015). We reasoned that T4 DNA polymerase could fill in those overhangs, creating 5–8 nucleotides insertions. Because the Cas12a cleaves the target DNA at the distal end of PAM (18–25 bp away), Cas12a can recut the allele until an indel produced upstream of the insertions to prevent recutting (Appendix Fig. S2H). Thus, we co-expressed T4 DNA polymerase along with Cas12a and guide RNA-Lb1 into cells and analyzed the proportion of alleles containing 5–8 nucleotides insertions. As expected, T4-WT and T4-D219A substantially increased the frequency of alleles carrying 5–8 nucleotide insertions compared to Cas12a-only treatment (Appendix Fig. S2I,2J). Hence, with Cas12a, T4 DNA polymerase was also capable of filling in the 5′

staggered ends with 5–8-bp overhangs, which is consistent with our results in Cas9-mediated editing.

## Combining Cas9 variants with T4 DNA polymerase expands the target range for CasPlus editing

CasPlus editing requires Cas9 to generate DNA ends with 5′ overhangs. Certain Cas9 variants preferentially produce staggered ends with 1-bp overhangs, while wild-type Cas9 (Cas9-WT) mainly generates blunt ends at cleavage sites (Jiang et al, 2016; Shou et al, 2018). We hypothesized that combining these Cas9 variants with T4 DNA polymerase would favor 1-bp insertions at target sites (Fig. 2D,2E). At TS11, the Cas9 variants F916P, F916del, R919P, and Q920P produced an average of 15% 1-bp insertions, whereas Cas9-WT produced <5% 1-bp insertions (Fig. 2F). Combining these Cas9 variants with T4-WT resulted in an average of ~50% 1-bp insertions and ~4.5% deletions whereas combining Cas9-WT and T4-WT yielded ~15% 1-bp insertions and ~59% deletions (Fig. 2G). We tested five target sites that rarely induced 1-bp insertions by Cas9-WT with or without T4-WT. Cas9-F916P led to an average 4.3-fold and Cas9-F916del led to an average 5.1-fold increase in 1-bp insertions at the expense of deletions across the five target sites with T4-D219A compared to Cas9-WT or Cas9 variants alone (Appendix Fig. S2K). Our strategy expanded targetable sites for CasPlus editing for precise insertions. We predicted that Cas9 variants with T4 DNA polymerase could produce longer insertions (2–4-bp) at target sites where Cas9-WT and T4-WT only produce 1–3-bp insertions (Appendix Fig. S2L). With T4-WT, Cas9 variants F916P, F916del, or Q920P substantially increased 3-bp insertions at tdTomato-d151A site where Cas9-WT produced 2-bp insertions (Appendix Fig. S2M,2N). With T4-WT or T4-D219A, Cas9-F916P and Cas9-F916del promoted 2-bp insertions at TS5, 3-bp insertions at TS17, and 4-bp insertions at TS18 whereas Cas9-WT produced 1-bp insertions at TS5, 2-bp insertions at TS17 and 3-bp insertions at TS18 (Appendix Fig. S2O).

The four versions of the CasPlus system (Fig. 2H) generate distinct editing outcomes. At target sites where CasPlus-V1 predominantly favors 1–3-bp insertions, CasPlus-V2 offers even higher efficiency. CasPlus-V3 and -V4 can increase insertion length by 1-bp. At target sites where CasPlus-V1 mainly generates 1- or 2-bp deletions, CasPlus-V2, -V3, and -V4 favor 1-bp insertions (Appendix Fig. S2P).

## CasPlus enhanced the efficiency of *DMD* (exon 52 deletion) correction

CasPlus editing enhances 1–2-bp insertions, which can efficiently correct diverse disease-causing frameshift mutations, such as those

found in *DMD* mutations with deleted exons. Precise insertion of 1 bp at the 3′ end of exon 51 or the 5′ end of exon 53 could reframe mutated *DMD* with exon 52 deleted (Fig. 3A). We generated all gRNAs targeted exon 51 or exon 53 which can potentially reframe *DMD* del exon 52 and determined their editing efficiencies in HEK293T cells (Fig. 3B). The gRNAs G10 and Ex51-G2 targeting exon 51 and G9 and Ex53-G3 targeting exon 53 had higher editing efficiencies compared to other gRNAs (Fig. 3C). Consistent with gRNA G10 and G9 (Appendix Fig. S2B), Ex51-G2 and Ex53-G3, when used with CasPlus-V1 or -V2, resulted in a 1.6-to-2-fold increase in correction efficiency compared to Cas9-only samples, primarily due to the increased 1-bp insertions (Appendix Fig. S3A). Compared to Cas9, CasPlus editing is more flexible and efficient in correcting *DMD* mutations. Using CasPlus-V1 and -V2, gRNA G10 led to a 4.3-to-6.5-fold increased correction efficiency in exon 51, and gRNA G9 increased correction efficiency by 2.1-to-2.5-fold in exon 53 versus using Cas9-only in a human induced pluripotent stem cell (iPSC) line with DMD exon 52 deletion (DMD-del52). (Fig. 3D; Appendix Fig. S3B). We differentiated the pool of edited iPSCs and a single clone (SC) with 1-bp insertion into cardiomyocytes (iCMs). Compared to Cas9 editing, frequencies of in-frame mRNAs ($3n + 1$) in iCMs resulting from CasPlus-V1 and -V2 editing were ~2.5-fold higher in exon 51 and ~1.5-fold higher in exon 53 (Fig. 3E). Western blot analysis confirmed that CasPlus-V1 and -V2 editing produced 1.5-to-3-fold higher dystrophin expression than Cas9-only editing (Fig. 3F).

To investigate whether co-expression of T4 DNA polymerase and Cas9 increases genome-wide off-target effects, we performed whole-genome sequencing (WGS) in HEK293T cells with or without Cas9 or CasPlus editing using gRNA G10 (Appendix Table S3). We observed no apparent differences in insertion/deletion (indel) or single-nucleotide polymorphism (SNP) profiles when comparing CasPlus to Cas9 editing (Appendix Fig. S3C–3F; Appendix Table S4). We did not detect gene editing in Cas9- or CasPlus-edited cells at off-target sites predicted by Cas-OFFinder (Bae et al, 2014) (Appendix Fig. S3G,3H; Appendix Table S5). Phage T4 DNA polymerase is essential for replication of phage genome (De Waard et al, 1965). To assess the impact of transient co-expression of engineered T4 DNA polymerase and Cas9 on cell cycle progression, we stained the transfected HEK293T cells using propidium iodide (PI) and measured the distribution of cells in three major phases of the cycle (G1, S, and G2/M) via flow cytometry. A slight but not significant increase of cells in S phase was observed in cells with CasPlus-V1 but not CasPlus-V2 editing verse Cas9 editing. Collectively, these results indicate that CasPlus potentially did not raise additional safety concerns compared to Cas9 editing (Appendix Fig. S3I).

## Repression of on-target large deletions by CasPlus editing in iPSCs

Unexpected on-target large deletions can arise from the long-range end resection that occurs during Cas9 editing (Yoo et al, 2022) (Fig. 4A). In our corrected DMD-del52 iCMs, at the mRNA level, we detected the unexpected skipping of the whole exon 51 using gRNA G10 and exon 53 using gRNA G9 in Cas9-only editing (Appendix Fig. S4A,4B). We speculated that these exons loss resulted from unexpected on-target large deletions which eliminate exon 51 or exon 53. To investigate this, we PCR-amplified a ~ 2-

kilobase (kb) region in pools of edited cells (Appendix Fig. S4C). We observed several lower bands representing deletions of ~0.5 kb distal to gRNA G10 and ~1.2 kb distal to gRNA G9 in Cas9-only but not CasPlus-edited iPSCs and iCMs (Fig. 4B; Appendix Fig. S4D–4G). We amplified a ~5-kb region around the *DMD* exon 51 and 53 target sites from pools of edited iPSCs and sequenced the PCR amplicons using PacBio sequencing technology. Up to 23.0% of the PacBio reads contained deletions of 200–3000 bp around the exon 51 cleavage site in Cas9-edited cells (Fig. 4C; Appendix Fig. S4H; Appendix Table S6). This on-target effect was not observed in untreated cells or cells edited with CasPlus-V1 or -V2. In untreated cells, we detected ~3-kb deletions around *DMD* exon 53 in 13.2% of the PacBio reads. This result likely resulted from an artifact due to PCR amplification process, as 3-kb deletions of similar scale were observed in all tested cells [Cas9 (11.1%); CasPlus-V1 (9.4%); CasPlus-V2 (14.8%)]. On *DMD* exon 53, reads with deletions of 200–3,500 bp around the cut site were 2.8- to 5.1-fold less frequent in cells edited with CasPlus-V1 (9.5%) and -V2 (17.4%) versus Cas9 (48.9%) (Fig. 4D; Appendix Fig. S4H; Appendix Table S6). We genotyped single clones sorted from edited iPSC pools to confirm our results. Three of 33 clones (9.1%) using Cas9 with gRNA G10 and three of 13 clones (23.1%) using Cas9 with gRNA G9 contained deletions of 300 bp to 10 kb. No clones from CasPlus-V1- or CasPlus-V2-edited cells had deletions >150 bp (Fig. 4E; Appendix Fig. S4I–4K). To investigate whether CasPlus editing could inhibit on-target large deletions on other genomic loci, we used a flow cytometric assay (Kosicki et al, 2022; Kosicki et al, 2018) to detect and isolate cells containing large deletions and more complex rearrangements on the X-linked *PIGA* gene. We transfected male iPSCs with a gRNA targeted the intron 1 and observed a significant decrease in the frequency of PIGA⁻ populations in cells treated with CasPlus verse Cas9 (Appendix Fig. S4L). Genotyping of single clones from PIGA⁻ populations confirmed that all clones contained large deletions (>265 bp) resulting in partial or entire deletion of exon 2 (Appendix Fig. S4M). Thus, CasPlus efficiently represses on-target large deletions in mammalian cells.

## Repression of on-target large deletions by CasPlus editing in mouse germline

On-targe large deletions frequently occur during CRISPR/Cas9-mediated mouse embryo editing (Adikusuma et al, 2018; Papathanasiou et al, 2021). To evaluate the capacity of CasPlus editing in inhibiting on-target large deletions for mouse embryo editing, we microinjected Cas9 mRNA and gRNA against *Mybpc3* alone (Cas9) or combined with T4-WT (CasPlus-V1) or T4-D219A (CasPlus-V2) mRNA into fertilized mouse zygotes. Four days later, we harvested the embryo for future analysis (Fig. 5A). The editing efficiency of each embryo varied between 80 and 100% in both Cas9- and CasPlus-edited group (Appendix Fig. S5A). To assess the presence of large deletions, we performed multiple PCRs to amplify the proximal, distant, or both side of the PAM. Full-length and lower molecular mass PCR bands were analyzed by Sanger sequencing (Fig. 5B; Appendix Fig. S5B). The sequencing results showed that 8/10 embryos (80%) treated with Cas9, and 3/10 embryos (30%) treated with CasPlus-V1 or V2 harbored >500 bp large deletions (Fig. 5C,5D; Appendix Fig. S5C,5D). Collectively, these results indicated that CasPlus-V1- and -V2 editing efficiently repressed Cas9-mediated on-target large deletions in mouse germline editing.

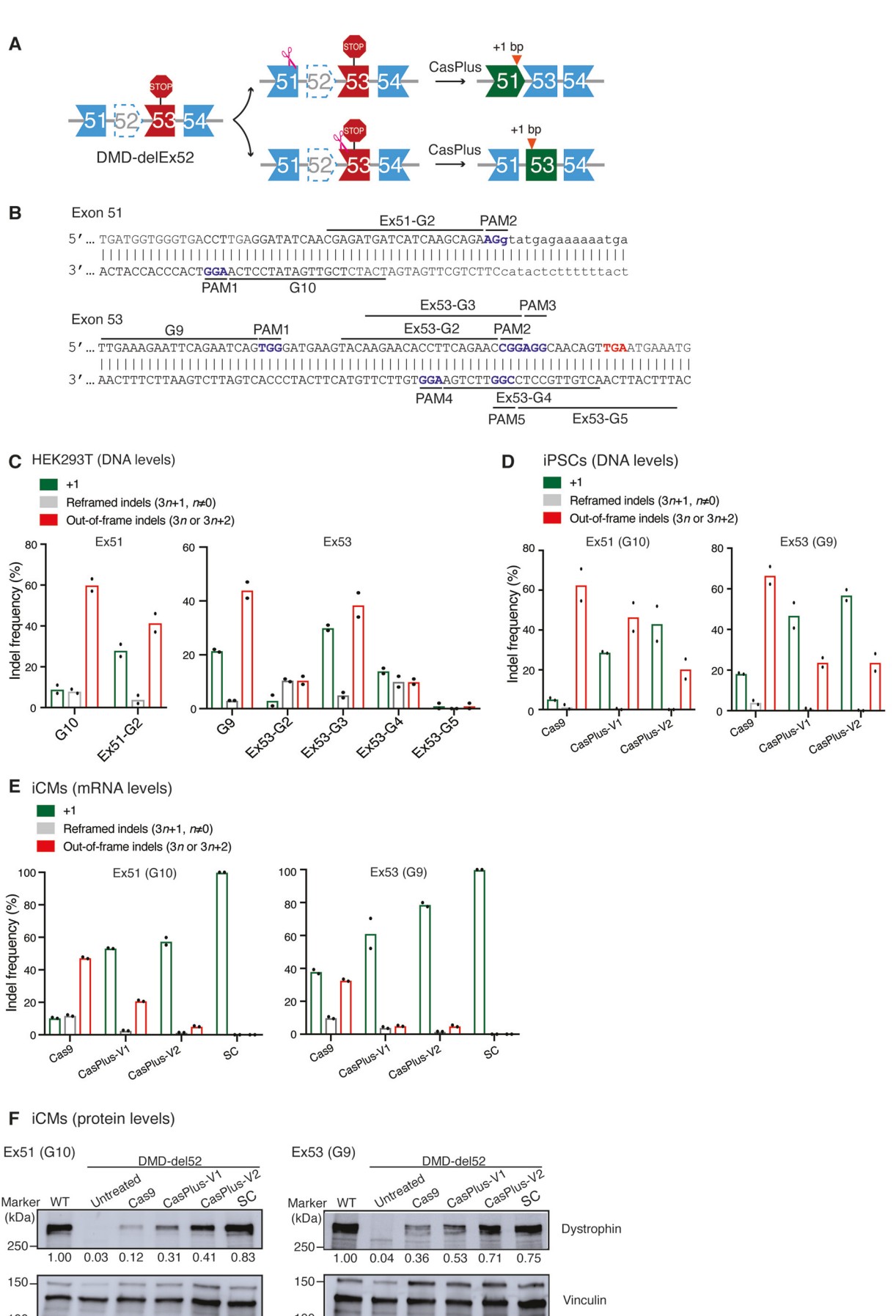

**Figure 3. CasPlus enhanced efficiency of *DMD* (exon 52 deletion) correction.**

(A) Deletion of *DMD* exon 52 creates a premature stop codon in exon 53, preventing dystrophin expression. Two CasPlus editing strategies can restore dystrophin expression by generating 1-bp insertions. (B) All potential gRNA sequences with an NGG PAM on *DMD* exons 51 and 53 available for restoring the *DMD* exon 52 deletion mutation. Red, premature stop codon. (C) Frequencies of in-frame ($3n + 1$) and out-of-frame ($3n$ or $3n + 2$) indels induced by Cas9 with gRNAs described in (B) in HEK293T cells. Frequencies were calculated based on ICE analysis on Sanger sequencing files. Each dot represents one biological replicate. (D) Frequencies of in-frame ($3n + 1$) and out-of-frame ($3n$ or $3n + 2$) indels induced by Cas9, CasPlus-V1, and CasPlus-V2 with gRNA G10 or G9 in DMD-del52 iPSCs. Each dot represents one biological replicate. (E) Edited DMD-del52 iPSCs described in (D) were differentiated into iCMs and then in-frame ($3n + 1$) and out-of-frame ($3n$ or $3n + 2$) indels on the mRNA levels were analyzed by HTS. Each dot represents one biological replicate. (F) Western blot analysis of the expression of dystrophin and vinculin in DMD-WT iCMs, untreated DMD-del52 iCMs, DMD-del52 iCMs differentiated from iPSCs with edits at *DMD* exon 51 (left) or 53 (right), or DMD-del52 single clone containing 1-bp insertions. The normalized dystrophin quantitation is shown below the gel image. Source data are available online for this figure.

## Repression of on-target chromosomal translocations during multiplex gene editing by CasPlus editing in primary T cells

Chromosomal translocations can occur when two simultaneous DSBs are present on two chromosomes during Cas9-mediated single-gene editing or multiple gene editing (Amit et al, 2021). To investigate whether CasPlus editing can reduce chromosomal translocations, we recapitulated translocation events between the genes *CD74* (chromosome 5) and *ROS1* (chromosome 6) in HEK293T cells (Choi and Meyerson, 2014) (Appendix Fig. S6A). We PCR-amplified the breakpoint junction regions on the fused chromosomes and assessed translocation efficiencies. The translocation frequencies between *CD74* and *ROS1* were ~fivefold lower with CasPlus-V1 and ~2-fold lower with CasPlus-V2 versus Cas9 editing (Appendix Fig. S6B,6C). The frequencies of insertions at *ROS1* and *CD74* individual sites were higher with CasPlus-V1 and -V2 editing compared to Cas9 (Appendix Fig. S6D). We observed similar trends of repression of chromosomal translocations in iPSCs (Appendix Fig. S6E–6G). Genotyping revealed that three of 95 clones (3.2%) from the Cas9-edited iPSC pool but no clones from the CasPlus-edited iPSC pools contained *ROS1-CD74* and *CD74-ROS1* translocations (Appendix Fig. S6H–6K).

In a clinical trial (NCT03399448), chromosomal translocations were frequently observed between targeted chromosomes in Cas9-engineered T cells when delivered with three gRNAs targeting *PDCD1*, *TRBC1/2*, and *TRAC* genes on different chromosomes (Stadtmauer et al, 2020). We compared the same three gRNAs with Cas9 or CasPlus to assess translocations in these three chromosomes in HEK293T cells (Fig. 6A; Appendix Fig. S6L). PCR demonstrated that CasPlus-V1 led to a 2.5-to-4.5-fold decrease in all translocation types in the three chromosomes (Fig. 6B,6C; Appendix Fig. S6M,6N). TaqMan assays revealed that CasPlus-V1 decreased an average 5.8-fold, and CasPlus-V2 decreased an average 3.9-fold translocations compared to Cas9 editing (Fig. 6D). CasPlus-V1 editing induced similar gene disruption efficiency to Cas9 editing for four genes (Fig. 6E). CasPlus-V2 induced a similar gene disruption efficiency but was less effective in repressing chromosomal translocations compared to CasPlus-V1. We recommended CasPlus-V1 to generate CAR-T cells.

We investigated CasPlus editing efficacy in suppressing chromosomal translocations among genes *PDCD1*, *TRBC1/2* and *TRAC* in primary human T cells. First, we nucleofected the activated primary human T cells with Cas9 mRNA and *PDCD1*, *TRBC1/2* and *TRAC* sgRNAs alone (Cas9) or in conjunction with T4-WT (CasPlus-V1) or T4-D219A (CasPlus-V2) mRNA. Four days after mRNA introduction, we accessed translocations using

PCR and TaqMan assay. CasPlus-V1 induced 1.5-to-11-fold fewer translocations than Cas9 editing in PCR assays (Fig. 6F,6G). TaqMan assays confirmed that CasPlus-V1 reduced translocations by 1.2-to-4.3-fold relative to Cas9 (Fig. 6H). As in HEK293T cells, CasPlus-V2 also suppressed translocations but was less efficient compared to CasPlus-V1. Analysis of the on-target gene editing through sequencing revealed that CasPlus editing had similar knockout efficiency at *PDCD1*, *TRBC1/2*, and *TRAC* sites in T cells (Fig. 6I). Together, our findings indicate that CasPlus editing significantly inhibits Cas9-mediated on-target chromosomal translocations and offers a potentially safer editing strategy to engineer T-cell and other ex vivo cell therapies.

## Discussion

Despite the successes achieved with all CRISPR/Cas-based genome editing technologies, including base editing and prime editing, the clinical potential of the technologies is currently constrained by the safety and efficacy concerns in cultured cells and animal models, and likely in humans. Unexpected on-target large deletions and chromosomal translocations from Cas9 editing are an emerging risk factors for in vitro, in vivo and ex vivo applications (Kosicki et al, 2018; Leibowitz et al, 2021; Nahmad et al, 2022a; Stadtmauer et al, 2020). Our results show that in cultured human cells, the CasPlus editing platform can substantially reduce deleterious on-target large deletions and chromosomal translocations while maintaining equal or higher efficiency of desired edits (Appendix Fig. S6O). Future research will assess the safety and efficacy of CasPlus in animal models using adeno-associated virus and lipid nanoparticles delivery systems.

CasPlus utilizes T4 DNA polymerase to fill in the staggered end created by Cas9 or Cas9 variants, producing small insertions and reducing relatively large deletions. Fusion of DNA polymerase I or KF to Cas9 could increase the frequency of 1-bp deletions over >1 deletions by counteracting the DNA resection process (Yoo et al, 2022). We did not observe similar results when co-expressing DNA polymerase I or KF with Cas9 at the tdTomato-d151A site, which may reflect a sequence-specific effect. In addition, fusionT4-L-Cas9 and Cas9-L-T4 increased relatively large deletions (<25 bp) at the PAM distal region at *CLCN5* site which is used in previous study. Although the mechanism underlying the divergent outcomes observed when T4 DNA polymerase was delivered in trans and in cis remains unclear, it is believed that the exonuclease domain within T4 DNA polymerase may have contributed to these differences. One potential explanation is that the fusion of T4 DNA polymerase to Cas9 may impact the binding or accessibility of

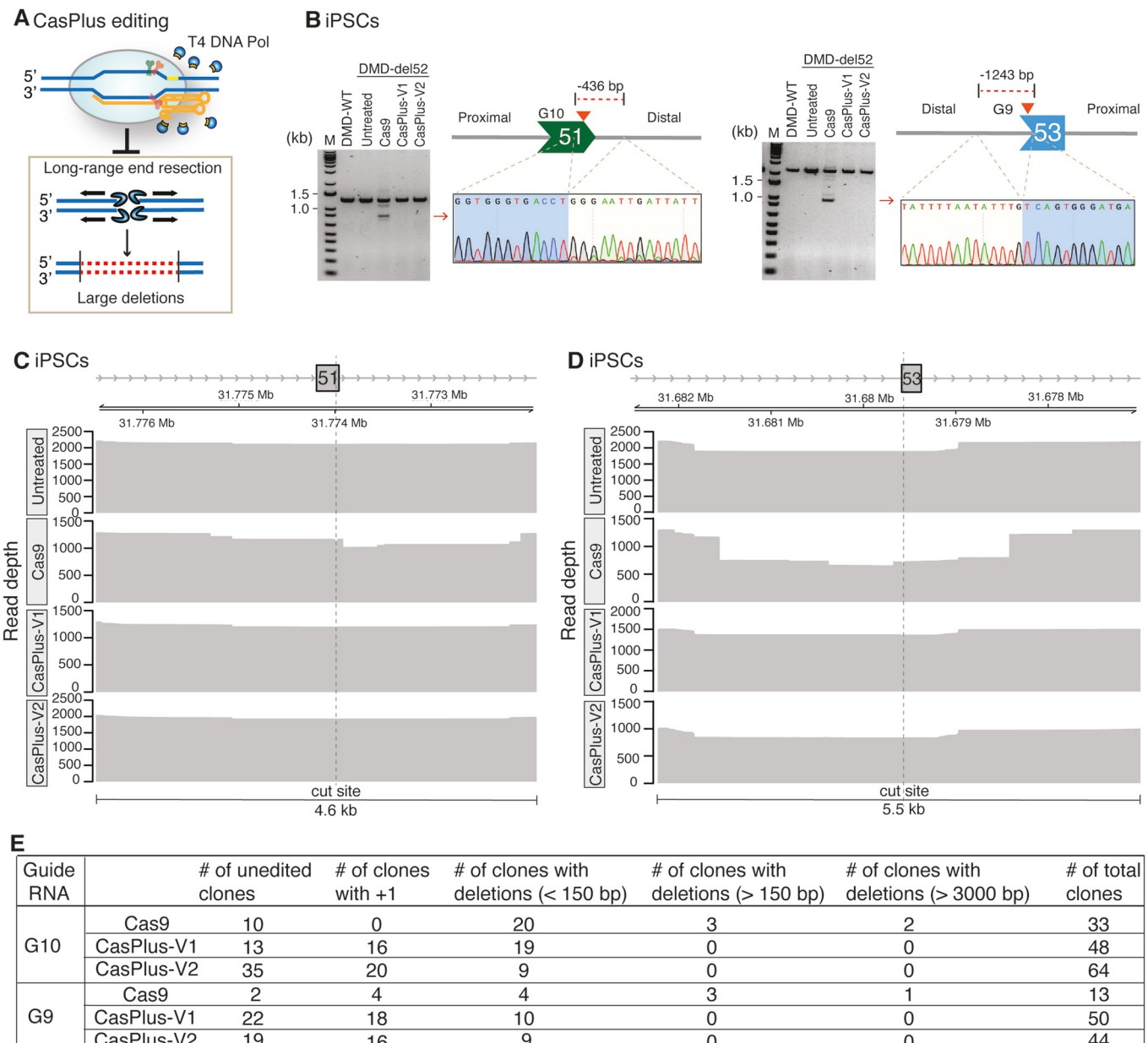

**Figure 4. Repression of on-target large deletions by CasPlus editing in iPSCs.**

(A) CasPlus editing represses the generation of large deletions by counteracting the long-term end-resection process. (B) Representative gel images showing the PCR products amplified from DMD-WT iPSCs, untreated DMD-del52 iPSCs or DMD-del52 iPSCs with edits at *DMD* exon 51 (left) or 53 (right). Lower bands (red arrowheads) were purified and sequenced. (C, D) Depth of PacBio reads at *DMD* exon 51 (C) or 53 (D) in untreated, Cas9-, CasPlus-V1-, and CasPlus-V2-edited DMD-del52 iPSCs. (E) Table summarizing the number of unedited single clones and single clones containing 1-bp insertions or deletions with different sizes. Single clones were isolated from Cas9-, CasPlus-V1-, and CasPlus-V2-edited DMD-del52 iPSCs. Source data are available online for this figure.

T4 DNA polymerase to DSBs. Further characterization of the T4 DNA polymerase, Cas9 and DNA crystal structure during DNA repair is necessary to elucidate this mechanism in more details.

Our data shows that CasPlus can favor templated 1–2-bp insertions as well as 1–2-bp deletions in a guide RNA context-dependent manner, but more comprehensive and systematic analysis is needed to predict or better control the outcomes of CasPlus editing. CasPlus-mediated 1–2-bp insertions could be harnessed to disrupt pathogenic genes or correct frameshift

mutations. Base editing and prime editing are versatile approaches to disrupt and correct genes without requiring exogenous templates and DSBs. Base editing can generate single-nucleotide mutations but not indels (Gaudelli et al, 2017; Komor et al, 2016). Prime editing can achieve single-nucleotide and indels editing (Anzalone et al, 2019). However, the efficiency of prime editing varies among different pegRNA constructs in diverse targets in cells and organoids and was averagely less than Cas9 when targeting the identical genomic sites (Geurts et al, 2021; Kim et al, 2021).

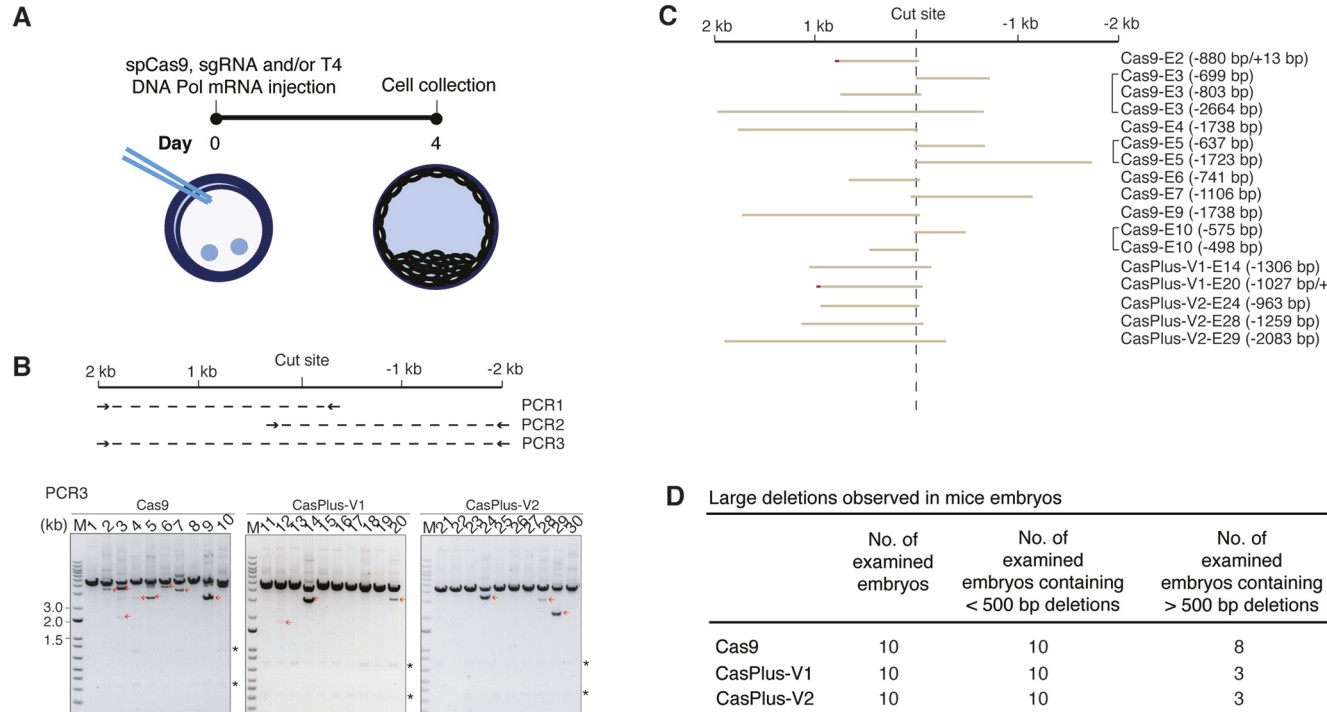

**Figure 5. Repression of on-target large deletions by CasPlus editing in mouse germline.**

(A) At day 0, Fertilized mouse zygotes were microinjected with Cas9 mRNA and a sgRNA alone or combined with T4 DNA polymerase mRNA. At day 4, all embryos were collected for analysis. (B) Top schematic illustrating the sizes and locations of the three PCR amplicons used to detect large deletions. Gel images showing the PCR result for PCR3. Small molecular mass products (red arrowheads) were sequenced. Asterisk, non-specific bands. (C) The sequencing results for the small molecular mass products observed in (B). (D) Table summarizing the total number of embryos tested, the number of tested embryos containing small deletions (<500 bp) and the number of tested embryos containing large deletions (>500 bp) in Cas9, CasPlus-V1- or -V2 edited group. Source data are available online for this figure.

Efficient prime editing can achieved by introducing a second nick on the non-edited stand, but can generate undesired DSBs and large deletions in mouse embryos (Tomomi Aida et al, 2020). Other methods can reduce translocations, including nuclease combinations (Bothmer et al, 2020) or Cas12f nuclease (Xin et al, 2022) for multiplex gene editing and fusion of Cas9 to an exonuclease to prevent repeat cleavage (Yin et al, 2022). However, unlike CasPlus, these approaches cannot precisely control the editing outcomes. Since CasPlus does not negatively affect the editing efficiency of regular Cas9 and can produces a dominant genotype of 1–2-bp indel with much fewer byproducts and on-target damage, it could be a safer and more efficient method optimized for human applications.

In addition, CasPlus' enhanced safety features can expand ex vivo applications, including chimeric antigen receptor T (CAR-T) cells to target cancer or autoimmune diseases. The US Food and Drug Administration placed temporary holds on CRISPR/Cas9-based clinical trials related to allogeneic CAR-T therapies following the detection of a chromosomal abnormality in a patient and requested control data on genomic rearrangement (Sheridan, 2022b). As eukaryotic DNA repair mechanisms are conserved, our data from HEK293T, iPSC, and primary T cells suggest that CasPlus can generate engineered cells with fewer chromosomal abnormalities for cell therapy applications. Hence, CasPlus may improve the safety and efficacy of gene editing to develop novel cellular therapies.

# Methods

## Plasmids

The vector pSpCas9(BB)-2A-GFP (PX458) (Addgene plasmid #48138) containing the human codon-optimized SpCas9 gene with 2A-GFP and the sgRNA backbone, vector pCMV-PE2 (Addgene plasmid # 132775), vector pCMV-BE3 (Addgene plasmids # 73021) were purchased from Addgene. p3xFlag-CMV-10 was a gift from Dr. Xiaodong Wang. pLentiV-SgRNA-tdTomato-P2A-BlasR was a gift from Dr. Lukas Dow. EF1A-CasRx-2A-EGFP was a gift from Dr. Patrick Hsu. Plasmids psPAX2 and pMD2G were gifts from Dr. Lei Bu. pBSU6_FE_Scaffold_rsv_GFP was a gift from Dr. Dirk Grimm. UCOE-SFFV-dCas9-BFP-KRAB was a gift from Dr. Richard Tsien. To construct pSpCas9(BB)-2A-BFP, the BFP gene was amplified from UCOE-SFFV-dCas9-BFP-KRAB and cloned into pSpCas9(BB)-2A-GFP (PX458) to replace GFP via Gibson assembly. To construct the lentiviral vector expressing tdTomato-d151A, the tdTomato-d151A gene was synthesized by Integrated DNA Technologies (IDT). First, it was cloned into vector p3xFlag-CMV-10, then the CMV-10-tdtomato-d151A was cloned into pLentiv-SgRNA-tdTomato-P2A-BlasR using MluI and BamHI restriction sites. To construct For DNA polymerase cloning, the coding sequences of DNA polymerase 4, DNA polymerase I, Klenow fragment, T4 DNA polymerase, RB69 DNA polymerase, and T7 DNA polymerase were codon-optimized for human cell

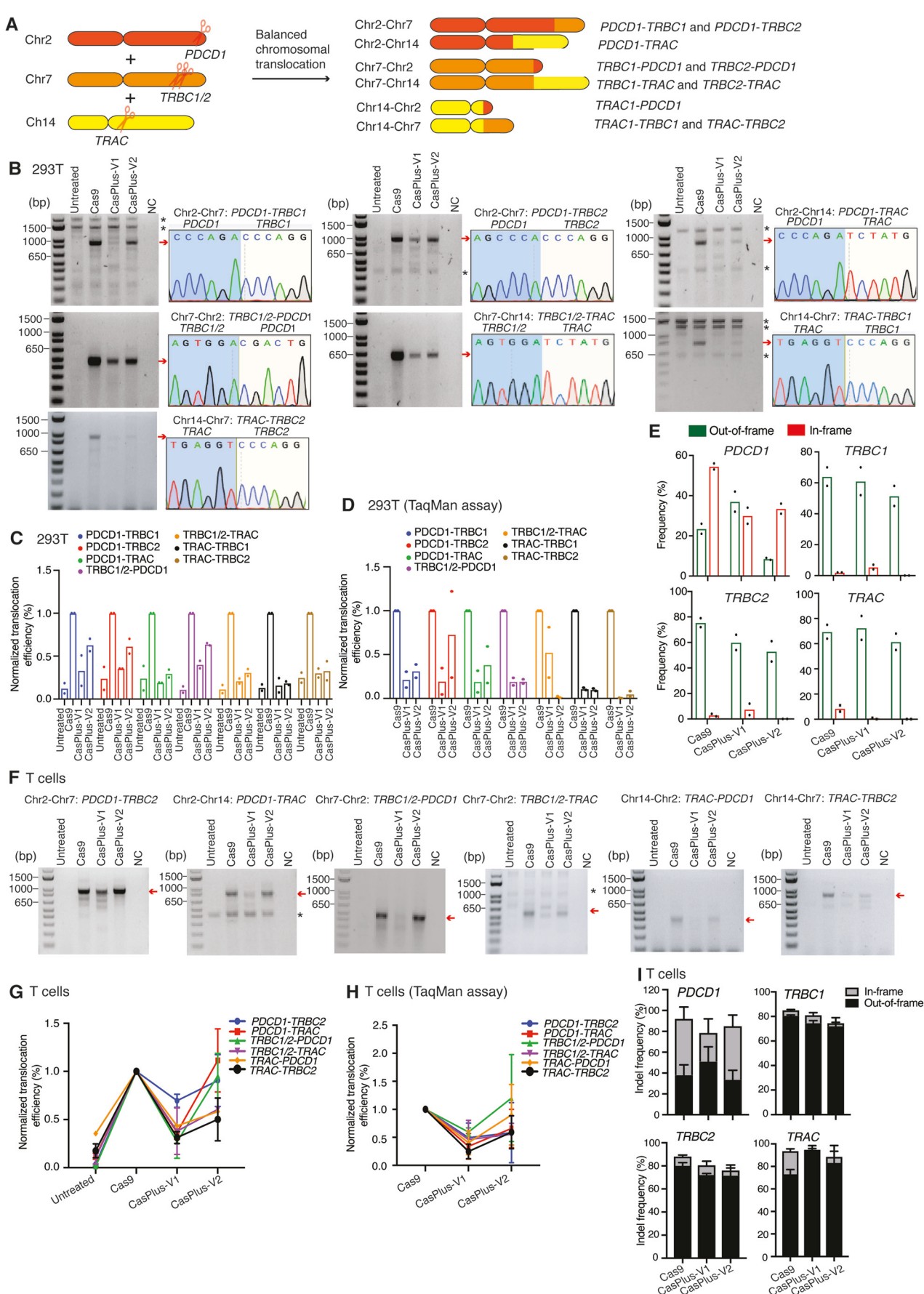

**Figure 6.  Repression of on-target chromosomal translocations during multiplex gene editing by CasPlus editing in primary T cells.**

(A) Schematic illustrating the balanced translocations among genes *PDCD1*, *TRBC1/2*, and *TRAC*. (B) Representative gel images demonstrating the balanced translocations detected in HEK293T cells during Cas9, CasPlus-V1, or CasPlus-V2 editing (see "Methods"). Balanced translocation of Chr14-Chr2: *TRAC-PDCD1* was undetectable by PCR. Bands with expected size (red arrowheads) were purified, TA-cloned and sequenced (one of the TA clone results was show for each type of translocation). Asterisk, non-specific bands. (C) Normalized quantification of data in (B). Each dot represents one biological replicate. (D) Evaluation of balanced chromosomal translocations in HEK293T cells during Cas9, CasPlus-V1, or CasPlus-V2 editing by TaqMan assay (see "Methods"). For each assay, the translocation efficiency in Cas9-edited cells is set as 1. Each dot represents one biological replicate. (E) Frequencies of out-of-frame and in-frame indels at four individual sites in HEK293T cells during Cas9, CasPlus-V1, or CasPlus-V2 editing. Frequencies were calculated based on ICE analysis on Sanger sequencing files. Each dot represents one biological replicate. (F) Representative gel images demonstrating the balanced translocations detected in primary human T cells during Cas9, CasPlus-V1, or CasPlus-V2 editing (see "Methods"). Balanced translocation of Chr14-Chr7: *TRAC-TRBC1* and Chr2-Chr7: *PDCD1-TRBC1* was undetectable by PCR. (G) Normalized quantification of data in (F). Values and error bars reflect mean ± SEM of *n* = 3 replicates from two donors. (H) Evaluation of balanced chromosomal translocations in primary human T cells during Cas9, CasPlus-V1, or CasPlus-V2 editing by TaqMan assay (see "Methods"). Values and error bars reflect mean ± SEM of *n* = 3 replicates from two donors. (I) Frequencies of out-of-frame and in-frame indels at four individual sites in primary human T cells during Cas9, CasPlus-V1, or CasPlus-V2 editing. Frequencies were calculated based on ICE analysis on Sanger sequencing files. Values and error bars reflect mean ± SEM of *n* = 3 replicates from two donors. Source data are available online for this figure.

expression using the Genewiz Codon Optimization tool. For each DNA polymerase, an expression cassette containing the polymerase, a MCP (MS2 bacteriophage coat protein) and a hemagglutinin (HA) tag, two copies of a nuclear localization sequence (NLS), and a flexible linker (L) were synthesized from Genewiz and cloned into EF1A-CasRx-2A-EGFP via Gibson assembly. Mutations of T4 DNA polymerase and RB69 DNA polymerase were introduced into the vectors EF1A-MCP-T4 DNA-Polymerase-2A-EGFP and EF1A-MCP-RB69 DNA-polymerase-2A-EGFP, respectively, via Gibson assembly. To construct Cas9 and T4 DNA polymerase fusion protein, T4 DNA polymerase was amplified from EF1A-MCP-T4-Pol-2A-GFP and cloned into backbone pSpCas9(BB)-2A-GFP (PX458) via Gibson assembly. Mutations of Cas9 were generated in the backbone pSpCas9(BB)-2A-GFP (PX458) via Gibson assembly. To test CasPlus editing, Guide RNAs were cloned into pSpCas9(BB)-2A-GFP (PX458) or Cas9 variants derived from pSpCas9(BB)-2A-GFP (PX458) according to the CRIPSR plasmid instructions from the Feng Zhang Lab (Ran et al, 2013). Prime editing guide RNAs and guide RNAs for creating a second nick were cloned into pBSU6_FE_Scaffold_rsv_GFP via Gibson assembly or restriction digestion and ligation. All guide RNA sequences are listed in Appendix Table S8. All DNA sequences synthesized for vector constructions are listed in Appendix Table S9.

### Generation of HEK293T cell lines containing the tdTomato-d151A reporter gene

To generate a stable tdTomato-d151A reporter cell line in HEK293T (ATCC, CRL-3216) cells, we co-transfected pLentiV vector expressing tdTomato-d151A, the lentiviral helper plasmids psPAX2 and pMD2G, and pEGFP (Lonza) into HEK293T cells. Single cells expressing GFP were isolated in 96-well plates 72 h post-transfection and genotyped 2 weeks later. Positive clones were then stored and expanded for subsequent experiments. Primers for genotyping are shown in Appendix Table S7.

### Generation of male iPS cell lines containing the DMD exon 52 deletion

Male wild-type iPSCs (NCRM-1) were electroporated with vectors expressing Cas9, GFP, and a pair of guide RNAs specific for the deletion (DMD-Ex52-g1 and DMD-Ex52-g2, see Appendix Table S8). Single cells expressing GFP were isolated in 96-well plates 72 h post-transfection and genotyped 2 weeks later. Positive clones containing the *DMD* exon 52 deletion were stored and expanded

for subsequent experiments. Primers for genotyping are shown in Appendix Table S7.

### Transfection and sorting of HEK293T cells

HEK293T cells were transfected using Lipofectamine 2000 Transfection Reagent (ThermoFisher Scientific) according to the manufacturer's instructions. Cell sorting was performed by the Cytometry & Cell Sorting Laboratory Core Facility at New York University Langone Health. Briefly, HEK293T cells were seeded into 12-well plates. After 20–24 h, cells were co-transfected with 1 µg vectors expressing Cas9 (with 2A-GFP) and a sgRNA, and 1 µg vectors expressing one of the DNA polymerases. Seventy-two hours post-transfection, cells were dissociated using a trypsin-EDTA solution (Corning) for 2 min at 37 °C. Subsequently, 2 ml of warm Dulbecco's modified Eagle's medium (DMEM) (Corning) supplemented with 10% fetal bovine serum (FBS) (Gemini Bio-Products) was added. The resuspended cells were transferred into a 15-ml Falcon tube and centrifuged at 1000 rpm for 5 min at room temperature. The medium was then removed, and the cells were resuspended in 0.4–1 ml DMEM. Cells were filtered through the 50-µm-mesh cap of a CellTrix strainer (Sysmex). Cells expressing GFP were sorted by flow cytometry into a 5-ml polypropylene round-bottom tube (Corning) for immediate DNA extraction.

### Isolation of raw DNA from sorted cells

Protease K (20 mg/ml) (Qiagen) was added to DirectPCR Lysis Reagent (Viagen Biotech Inc.) to a final concentration of 50 µg/ml. Sorted cells ($4 \times 10^4$–$1 \times 10^5$) were centrifuged at 4 °C at 12,000 rpm for 5 min and the supernatant was discarded. Cell pellets were resuspended in 20–50 µl of DirectPCR/protease K solution, incubated at 55 °C for >2 h or until no clumps were observed, incubated at 85 °C for 30 min, and then spin down briefly (10 s). In total, 1–2 µl DNA was used for PCR amplification. All PCR primer sequences are summarized in Appendix Table S7.

### Prime editing

HEK293T cells were co-transfected with 1 µg vectors expressing PE2, 1 µg vectors expressing pegRNAs, and 0.5 µg vectors expressing a second nicking sgRNA. Cells expressing GFP were sorted, and DNA was extracted from sorted cells as described above. PegRNAs and sgRNAs for prime editing are listed in Appendix Table S2.

### Human iPSC maintenance and nucleofection

Human iPSC lines were cultured in Stemflex™ medium (Thermo-Fisher) and passaged approximately every 3 days (1:8–1:12 split ratio). One hour before nucleofection, iPSCs were treated with 10 μM ROCK inhibitor (Y-27632) and dissociated into single cells using Accutase (Innovative Cell Technologies Inc.). Cells ($8 \times 10^5$) were mixed with 2 μg of a vector expressing Cas9 (with 2A-GFP), and a guide RNA, as well as 2 μg of a vector encoding a DNA polymerase. This mixture was electroporated into cells using the P3 Primary Cell 4D-Nucleofector X kit (Lonza) according to the manufacturer's protocol. After nucleofection, iPSCs were cultured in StemFlex™ medium supplemented with CloneR (10×) (STEM-CELL Technologies) and antibiotic-antimycotic (100×) (Thermo-Fisher). Three days after nucleofection, cells expressing GFP were sorted as described above and replated in StemFlex™ medium. Ten to fifteen days after sorting, cells were harvested for DNA isolation.

### Cardiomyocyte differentiation and purification

Human iPSCs (edited iPSC pools or single clones with 1-bp insertions) were induced for differentiation into cardiomyocytes according to the manufacturer's instructions using the PSC Cardiomyocyte Differentiation Kit (ThermoFisher Scientific). At 15–20 days after differentiation initiation, cells were purified in RPMI-1640 medium lacking glucose supplemented with B27 (ThermoFisher Scientific). Cells were cultured in this medium for 2–4 days. Cardiomyocytes were used for experiments on days 30–40 after the initiation of differentiation.

### RNA extraction and cDNA synthesis

RNA from iPSC-derived cardiomyocytes was extracted using TRIzol (ThermoFisher Scientific) according to the manufacturer's protocol. cDNA was synthesized using the Superscript III First-Strand cDNA Synthesis Kit (ThermoFisher Scientific) according to the manufacturer's instructions. All RT-PCR primer sequences are summarized in Appendix Table S7.

### Cell cycle analysis

HEK293T cells were transfected with Cas9 (T2A-GFP) and gRNA G10, or Cas9 (T2A-GFP), gRNA G10 and T4-WT or T4-D219A. Cells were harvested 24 h post-transfection, washed in PBS, and fixed using cold 70% ethanal/PBS overnight at 4 °C. Cells were washed with PBS and then incubated with PBS containing RNase A (1 mg/ml) at room temperature for an hour. Cells were centrifuged and resuspended in PBS containing RNase A (1 mg/ml) and PI (0.5 μg/ml) at room temperature for 15 min. Following PI staining, cells were transferred to 5-ml polypropylene round-bottom tube for cell cycle analysis in Bio-Rad ZE5 analyzer.

### Western blotting

HEK293T cells and cardiomyocytes (iCMs) differentiated from iPSCs were harvested, centrifuged, and lysed with RIPA lysis buffer (Santa Cruz Biotechnology) according to the manufacturer's protocol. Samples were lysed and centrifuged, and the supernatant was incubated at 95 °C for 10 min in the presence of Laemmli sample buffer (Bio-Rad). Proteins (20 μg per sample) were separated on Mini-PROTEAN TGX 4–15% precast SDS-PAGE gels (Bio-Rad) for 1–3 h at 100 V and then transferred to PVDF membrane (Bio-Rad) at 40 V for 2–8 h. Membranes were

blocked with 5% non-fat milk (Santa Cruz) for 1–2 h. Membranes were probed overnight at 4 °C with anti-HA antibody (MBL, M180-3) and anti-glyceraldehyde-3-phosphate dehydrogenase antibody (Sigma-Aldrich, G8795) or with anti-dystrophin (Sigma-Aldrich, D8168) and anti-vinculin antibody (Sigma-Aldrich, V9131). Membranes were then washed, probed with a goat anti-mouse or goat anti-rabbit IgG H + L-HRP conjugated secondary antibody (1:5000-1:10,000) (Bio-Rad) for 1 h, and visualized with Luminol reagent (Santa Cruz) according to the manufacturer's protocol.

### Detection of large deletions in iPSCs

DMD-del52 iPSCs were co-electroporated with 2 μg vectors expressing Cas9 (with 2A-GFP), and G10 or G9 and either 2 μg empty vectors or vectors expressing T4-WT or T4-D219A. Cells expressing GFP were then sorted into 5-ml polypropylene round-bottom tubes 72 h post-electroporation. Bulk-sorted cells were replaced and expanded. DNA was isolated from expanded bulk cells using the DNeasy Blood and Tissue Kit (Qiagen) 2 weeks later and subjected to large deletions detection (PCR and PacBio sequencing). Single cells were isolated from bulk-edited cells into 96-well plates 2 weeks after electroporation and genotyped 2 weeks after isolation. Single cells containing one insert of G at *DMD* exon 51 or T at *DMD* exon 53 were stored and expanded for subsequent experiments. Bulk-edited iPSCs, and the single clones containing 1-bp insertions were further differentiated into iCMs. DNA was isolated from iCMs and subjected to large deletions detection. Primers for large deletions detection are summarized in Appendix Table S7.

### FLAER staining

Nucleofection in iPSCs with a gRNA against gene *PIGA* was performed as described above. Seventy-two hours after nucleofection, cells expressing GFP were sorted and expanded. FLAER staining was performed 1 week after expansion. Briefly, cells ($3 \times 10^5$) were harvested and resuspended in 0.1% bovine serum albumin (BSA) in PBS containing 1 μg/ml Cederlane FLAER (Alexa 488 proaerolysin variant) (Fisher Scientific). Cells were cultured in a rotator for 30 min at room temperature and then washed twice with 0.1% BSA in PBS for 2 min. Cells were resuspended with 0.1% BSA in PBS and analyzed in Bio-Rad ZE5 analyzer.

## In vitro transcription of T4 DNA polymerase

The Cas9 mRNA (5meC, Ψ) was purchased from TriLink Biotechnologies (L-6125) and single RNA against genes *PDCD1*, *TRAC*, *TRBC* or *Mybpc3* was synthesized from Synthego (https://ice.synthego.com/#/). To construct the vector for in vitro transcription of T4 DNA polymerase, the T4 DNA polymerase expression cassette (without EGFP) was amplified from EF1A-MCP-T4 DNA-Polymerase-2A-EGFP and cloned into pCMV-BE3 via Gibson assembly. The construct pCMV-T7-T4 DNA-polymerase were then used as template for in vitro transcription using Invitrogen™ mMESSAGE mMACHINE™ T7 ULTRA Transcription Kit (ThermoFisher Scientific). The RNA transcripts were purified by MEGAclear Transcription Clean Up Kit (ThermoFisher Scientific) and eluted with nuclease-free water (Ambion). The concentration of T4 DNA polymerase was measured by a NanoDrop instrument (Thermo Scientific).

### In vitro fertilization and microinjection

All animal procedures were approved by the Institutional Animal Care and Use Committee at the NYU Langone Medical Center. hDMDdel52/*mdx* female mice (Gifts from Dr. Nicole Datson) at 4 weeks of age were used as oocyte donors for superovulation, which was performed by intraperitoneal injection of PMSG (5 IU, Sigma-Aldrich) and hCG hormone (5 IU, Sigma-Aldrich). hDMDdel52/*mdx* male mice (Gifts from Dr. Nicole Datson) at 6–8 weeks of age were used as sperm donors. Four hours after in vitro fertilization, total volume of 10 µl solutions containing complexes of Cas9 mRNA (100 ng/µl) and Mybpc3 sgRNA (50 ng/µl) alone or together with T4 mRNA (50 ng/µl) diluted with DEPC-treated injection buffer (0.25 mM EDTA, 10 mM Tris, pH 7.4) were injected into cytoplasm of the zygotes. After microinjection, embryos were cultured in microdrops of KSOM + AA containing D-glucose and phenol red (Millipore) under mineral oil at 37 °C for 4 days in a humidified atmosphere consisting of 5% $CO_2$ in air.

### Detection of chromosomal translocations by PCR in HEK293T cells

HEK293T cells were co-transfected with a vector expressing Cas9 (with 2A-GFP), and guide RNAs targeting either genes *ROS1* and *CD74* or genes *PDCD1*, *TRBC1/TRBC2*, and *TRAC* individually or along with an empty vector or vector expressing T4-WT or T4-D219A. Transfected cells expressing GFP were sorted into 5-ml polypropylene round-bottom tubes 72 h post-transfection and sorted cells ($1 \times 10^6$) were immediately subjected to DNA extraction. DNA was extracted using the DNeasy Blood and Tissue Kit (Qiagen) and 1 µl (50 ng/µl) DNA was used for each PCR reaction. Chromosomal translocations were detected by PCR using a GoTaq kit with primers specifically recognizing the breakpoint junction region of each fused chromosomes. All the guide RNAs used for translocations detection are summarized in Appendix Table S8. All the primers used for PCR are summarized at Appendix Table S7.

### Detection of chromosomal translocations by TaqMan qPCR assay

A panel of TaqMan qPCR assays were performed to detect the balanced chromosomal translocations among genes *PDCD1*, *TRBC1/TRBC2*, and *TRAC* in HEK293T cells and primary T cells. To increase the sensitivity and accuracy of TaqMan assays in HEK293T cells, purified genomic DNA (25 ng) was first amplified according to the manufacturer's instructions using REPLI-g Single Cell kit (Qiagen). Amplified genomic DNA was then diluted into ~800 ng/µl and 1 µl DNA was used for each qPCR reaction. In primary T cells, DNA was extracted from unedited or edited cells, diluted into ~10 ng/µl, and 2 µl DNA was used for each qPCR reaction. For each assay, a separate 20x FAM-labeled master mix was prepared with 1.8 µl forward primer (100 µM), 1.8 µl reverse primer (100 µM), 0.5 µl FAM-labeled probes (100 µM), and 8.1 µl $H_2O$. A qPCR reaction was prepared with 10 µl 2× TaqMan gene expression master mix (Applied Biosystems), 1 µl 20× FAM-labeled master mix, 1 µl amplified DNA (~800 ng/µl) and 8 µl $H_2O$. Assays were run in StepOne Real-time PCR systems (Applied Biosystems) using the following program: 2 min at 50 °C, 10 min at 95 °C and followed by 40 cycles of 20 s at 95 °C, 1 min at 60 °C. The primers and probes used for TaqMan assays are listed at Appendix Table S7.

### Primary human T-cell isolation and stimulation

Human peripheral blood mononuclear cells (PBMCs) were purchased from Lonza. Frozen PBMCs were thawed and cultured in X-vivo 15 medium (Lonza) with 5% human AB serum (Heat inactivated) (Valley Biomedical) for one day. Primary T cells were isolated from PBMCs using EasySep Human T Cells Isolation Kits (StemCell Technologies) according to the manufacturer's instructions. Immediately after isolation, primary T cells were activated and stimulated with a 1:1 ratio of anti-human CD3/CD28 magnetic Dynabeads (ThermoFisher) to cells in X-vivo 15 medium supplemented with 5% human AB serum (Heat inactivated), 5 ng/mL IL-7 (PeproTech), 5 ng/mL IL-15 (PeproTech), and 200 U/mL IL-2 (PeproTech). Two days after stimulation, magnetic beads were removed, and T cells were ready for nucleofection.

### T-cell nucleofection

Nucleofection in T cells were performed using P3 Primary Cell 4D-Nucleofector™ X Kit S (Lonza) according to the manufacturer's instructions with minor modifications. Briefly, T cells ($\sim 5–7 \times 10^5$) were collected and resuspended in 20 µl P3 buffer. 1 µg Cas9 mRNA and 3 µg sgRNAs (*TRAC*, *TRBC*, *PDCD1* sgRNA, 1 µg each) alone or together with 2 µg IVT T4 mRNA were added to the cells before nucleofection in a Lonza 4D-Nucleofector with pulse code EO-115. 80 µl X-vivo 15 medium with 5% human AB serum and 200 U/mL IL-2 was added to the nucleofected cells before a 15 min recovery at 37 °C. Nucleofected T cells were plated at a density of $\sim 0.5–1 \times 10^6$ cells/ml in X-vivo 15 medium supplemented with 5% human AB serum and 200 U/mL IL-2 in 48-well plates and replenished as needed to maintain a density of $10^6$ cells per ml. Four days after nucleofection, edited cells were harvested for analysis.

### PCR amplicon preparation for high-throughput sequencing

To prepare for high-throughput sequencing, PCR amplicons of ~300 bp were amplified using a GoTaq kit (Promega), separated on a 2% agarose gel, and purified with the MinElute Gel Extraction Kit (Qiagen). For each sample, gel-purified PCR product was barcoded with the Nextera Flex Prep HT kit according to the manufacturer's instructions and sequenced using the MiSeq paired-end 150-cycle format by the Genome Technology Center Core Facility at New York University Langone Health.

### PCR amplicon preparation for whole-genome sequencing

To prepare for whole-genome sequencing (WGS), HEK293T cells were co-transfected with a vector expressing only Cas9, T4-WT, or T4-D219A proteins; Cas9 in combination with G10; or Cas9 and either T4-WT or T4-D219A in combination with G10. All the vectors transfected expressed GFP. Seventy-two hours post-transfection, cells expressing GFP were sorted into 5-ml poly-propylene round-bottom tubes for immediate DNA isolation. Genomic DNA was extracted using the DNeasy Blood and Tissue Kit (Qiagen), barcoded with the PCR-free library prep kit according to the manufacturer's instructions, and sequenced using a S2 300-cycle flow cell v1.5 by the Genome Technology Center Core Facility at New York University Langone Health.

### PCR amplicon preparation for PacBio sequencing

To prepare samples for PacBio sequencing, genomic DNA was extracted from iPSCs using the DNeasy Blood and Tissue Kit. Barcodes were added to the target region via a two-step PCR reaction. The first-round PCR was performed using LA Taq DNA polymerase (Takara) according to the manufacturer's instructions. The first-round PCR amplified a 5-kb region around the target site

using target-specific primers tailed with universal forward and reverse sequences. The second round of PCR re-amplified and barcoded the first round of PCR products using universal, barcoded forward and reverse primers. The final barcoded PCR products were sequenced using the SMRT Cell (1 M v3 LR) platform by the Genome Technology Center Core Facility at New York University Langone Health. All primers used for PacBio sequencing are summarized in Appendix Table S7.

### High-throughput sequencing

To detect indels in the high-throughput sequencing data, unmapped paired-end amplicon high-throughput sequencing reads were used as inputs into the CRISPResso2 tool to quantify the frequency of editing events (Pinello et al, 2016). The tool was run with default parameters (https://github.com/pinellolab/CRISPResso2).

### PacBio sequencing

Raw PacBio data were demultiplexed with the corresponding barcode using the SMRTlink software to assign barcoded reads to each sample (smrtlink version: 8.0.0.80529, chemistry bundle: 8.0.0.778409, params: 8.0.0). Analysis of demultiplexed data was performed using PacBio tools distributed via Bioconda (https://github.com/PacificBiosciences/pbbioconda). For *DMD* exon 51 and 53 locus pileup, circular consensus sequences were converted to HiFi calls using the pbccs command and filtering for reads with support from at least three full-length subreads. The resulting fastq files were used as inputs to a custom python script that filtered for reads containing specific 50-bp index sequences at both the 5′ and 3′ regions of each read. Resulting filtered reads were mapped to the reference genome using minimap2 (ax splice --splice-flank=no -u no -G 5000). The genome coverage of the alignment files was calculated using the "bedtools genomecov -d" (v 2.27.1) command with all downstream analyses performed using custom R script (v4.1.1) and visualized with the Gviz1 package (Genomics S, 2016; Li, 2018). For *DMD* exon 51, the 5′ index sequence is tttttccaaacgtgctttcaggaaacagtggtctgcttgttgaagtctg and the 3′ index sequence is aatcctggaccagaggttccattgagctgagatcacaccattgcactcca. For *DMD* exon 53, the 5′ index sequence is ggactatattttgatttcatgttac aatcactagttttgtggggtctt and the 3′ index sequence is tgatgtgtattg ctgcagattcaatgtaagttcccgatacagataaagat.

### Genome-wide off-target analysis

FASTQ files were provided by the Genome Technology Center Core Facility. FASTQ files were aligned to human genome reference build GRCh38 using BWA-MEM aligner (v0.7.17) followed by the GATK (v4.2.1.0) best practices pipeline. The MarkDuplicatesSpark command was used to identify and mark PCR duplicate reads. BaseRecalibrator and ApplyBQSR commands were used along with known polymorphic sites to minimize systematic errors and improve downstream accuracy. Indel and SNP calling was done using Haplotypecaller, with the resulting file separated into SNPs and indels using the SelectVariants command. Variant filtering was performed using VariantFiltration with the following parameters: SNPs: $QD < 2.0$, $FS > 60.0$, $SOR > 4.0$, $MQ < 40.0$, $MQRankSum < -12.5$, ReadPosRankSum $< -8.0$; Indels: $QD < 2.0$, $FS > 200.0$, $SOR > 10.0$, $MQ < 40.0$. To ensure pipeline robustness, the site of editing in the Cas9 sample was identified and confirmed for editing using a custom parsing script and IGV (McKenna et al, 2010; Robinson et al, 2011).

### Predicted off-target analysis

Off-target events were predicted in silico using Cas-OFFinder allowing up to a 4-base mismatch with PAM sequences of either NGG or NAG (Bae et al, 2014). No off-target sites were predicted in this genome that were not found in WT samples.

## Statistical analysis

All samples used to test CasPlus editing were assayed in duplicate unless otherwise noted in the figure legends. All data were calculated based on HTS results unless otherwise noted in the figure legends. Data were presented as mean and standard error of the mean (SEM). Data were generated and statistical analysis was performed using GraphPad Prism software.

## Data availability

All data needed to evaluate the conclusions in the paper are present in the paper and/or the Supplementary information. Additional data related to this paper may be requested from the authors.

The source data of this paper are collected in the following database record: biostudies:S-SCDT-10_1038-S44318-024-00158-6.

## Peer review information

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

## Acknowledgements

The authors thank C Zhao, M Khurram, and Y Xiao for the cloning and sequencing of guide RNAs and plasmids; X Wang, L Dow, P Hsu, D Grimm, and L Bu for subcloning or lentivirus packaging vector plasmids; H Stower and R Barr for manuscript editing; A Namboodiripad, R Iyer, M Huang, and U. Andréo for comments and suggestions. This work was supported by grants from the Departmental Start-Up Grant, NYU Langone Health, and Kids Connect Charitable Fund.

## Author contributions

**Qiaoyan Yang**: Conceptualization; Resources; Data curation; Software; Formal analysis; Validation; Investigation; Visualization; Methodology; Writing—original draft; Writing—review and editing. **Jonathan S Abebe**: Software; Methodology; Writing—original draft. **Michelle Mai**: Data curation. **Gabriella Rudy**: Data curation; Writing—original draft. **Sang Y Kim**: Data curation; Writing—original draft. **Orrin Devinsky**: Conceptualization; Supervision; Funding acquisition; Methodology; Writing—original draft; Writing—review and editing. **Chengzu Long**: Conceptualization; Resources; Data curation; Software; Formal analysis; Supervision; Funding acquisition; Validation; Investigation; Visualization; Methodology; Writing—original draft; Project administration; Writing—review and editing.

Source data underlying figure panels in this paper may have individual authorship assigned. Where available, figure panel/source data authorship is listed in the following database record: biostudies:S-SCDT-10_1038-S44318-024-00158-6.

## Disclosure and competing interests statement

CL and OD are co-founders of Script Biosciences. CL and QY are listed on two patents related to this work (U.S. Application No. 63/335,625 and No. 63/109,909). The remaining authors declare no competing interests.

