## [Peer Review File · The EMBO Journal]

T4 DNA polymerase prevents deleterious on-target DNA damage and enhances precise CRISPR editing.

Chengzu Long, Qiaoyan Yang, Jonathan Abebe, Michelle Mai, Gabriella Rudy, Sang Kim, and Orrin Devinsky

Corresponding author(s): Chengzu Long (Chengzu.Long@nyulangone.org)

Review Timeline:

Submission Date:	2nd Jul 23
Editorial Decision:	12th Sep 23
Revision Received:	30th Oct 23
Editorial Decision:	9th Nov 23
Revision Received:	29th Nov 23
Editorial Decision:	24th Jan 24
Revision Received:	25th Mar 24
Editorial Decision:	31st May 24
Revision Received:	31st May 24
Accepted:	13th Jun 24

Editor: Cornelius Schneider

Transaction Report:

Dear Dr. Long,

Thank you for submitting your manuscript for consideration at the EMBO Journal. I again very much apologize for the delay in the peer review process.

Unfortunately, the last referee did not come back to me, and I have therefore decided to proceed further based on the comments from the remaining two reviewers, which are included below for your information.

As you will see from the reports, these reviewers find the use of T4 RNAP a useful and valuable addition to the CRISPR Cas9 arsenal and think that the study in general is well controlled. However, referee #2 raises several concerns which I find reasonable and productive that would need to be addressed before publication at The EMBO Journal.

Based on the interest expressed in the reports, I therefore invite you to address these issues in a revised version of the manuscript. If you have any additional questions, I am happy to discuss the revision in more detail via email or phone/videoconferencing. I should also add that it is The EMBO Journal policy to allow only a single major round of revision and that it is therefore important to resolve the main concerns at this stage.

We generally allow three months as standard revision time, which can be extended to six months in the case of major revisions. As a matter of policy, competing manuscripts published during this period will not negatively impact on our assessment of the conceptual advance presented by your study. However, please contact me as soon as possible upon publication of any related work to discuss the appropriate course of action. Should you foresee a problem in meeting this deadline, please let us know in advance to discuss an extension.

When preparing your letter of response to the referees' comments, please bear in mind that this will form part of the Review Process File and will therefore be available online to the community. For more details on our Transparent Editorial Process, please visit our website: <https://www.embopress.org/page/journal/14602075/authorguide#transparentprocess>. Please also see the attached instructions for further guidelines on preparation of the revised manuscript.

Please feel free to contact me if you have any further questions regarding the revision. Thank you for the opportunity to consider your work for publication. I look forward to discussing your revision.

Yours sincerely,

Cornelius Schneider

Cornelius Schneider, PhD
Editor
The EMBO Journal
c.schneider@embojournal.org

- a point-by-point response to the referees' comments, with a detailed description of the changes made (as a word file).

- a word file of the manuscript text.
 - individual production quality figure files (one file per figure)
 - a complete author checklist, which you can download from our author guidelines (<https://www.embopress.org/page/journal/14602075/authorguide>).
 - Expanded View files (replacing Supplementary Information)
- Please see out instructions to authors
<https://www.embopress.org/page/journal/14602075/authorguide#expandedview>

We realize that it is difficult to revise to a specific deadline. In the interest of protecting the conceptual advance provided by the work, we recommend a revision within 3 months (11th Dec 2023). Please discuss the revision progress ahead of this time with the editor if you require more time to complete the revisions. Use the link below to submit your revision:

Referee #1:

The authors study the use of various DNA polymerases to bias the outcome of CRISPR induced DSB repair towards nucleotide insertions rather than deletions in human cells. They find that the coexpression of Cas9 with T4 or RB69 DNA polymerase leads to strongly increased rates of Insertions and reduced frequencies of deletions and translocations. Using engineered versions of Cas9 and T4 Pol the Inertion frequencies can be further enhanced. This modified gene editing approach named CasPlus by the authors can be of great use for therapeutic reframing of mutant alleles (as demonstrated for DMD mutations) or the therapeutic knockout of modofifiers genes in CAR T cells while reducing undesried on target effects. Therefore this work will be of interest for the mammalian gene editing community in particular for therapeutic applications. The work will be well received by the audience of a journal that has a focus on gene editing, gene therapy or technical dvelopment.

Referee #2:

In this study, the researchers investigated the influence of different DNA polymerases on Cas9 and Cas12a genome editing outcomes, revealing that T4 DNA polymerase increases the occurrence of small insertions, while simultaneously reducing the frequency of large deletions and chromosomal translocations. Their approach was applied to diverse disease cell models, encompassing pluripotent stem cell-derived cardiomyocytes (DMD), mouse germline editing, and primary human T cells. The study is both elegant and comprehensive, and the data presented are abundant. This will be a useful contribution to the genome editing literature once the following issues are addressed.

Major points:

1. Supplementary Fig. 1 provides insight into the impact of Cas9 tethering on Pol function. While the authors speculate that the exonuclease domain, rather than the DNA polymerase domain, influences DSB repair and justifies this with strong deletions induced by tethered Cas9 and Pol, evidence supporting this claim remains thin. It would have been very straightforward to completely inactivate the polymerase domain, rather than use only the seven altered-fidelity mutants. This would be an important counterpart to the complete deletion of the exonuclease domain and is crucial to solidify their claim.
2. Supplementary Fig. 1g,h suggests that the orientation (Cas9-L-T4 vs. L-T4-Cas9) potentially impacts editing outcomes (deletions, etc.). Investigating whether T4 Pol fusion at the N or C terminus influences Cas9 cleavage efficiency, DSB repair, or both, would offer valuable insights. Furthermore, including a T4/Cas9 untethered group for comparison in Supplementary Fig. 1g,h would enhance clarity.
3. It would be valuable to explore potential cytotoxic effects of exogenous T4 polymerase and its influence on the cell cycle. Did the authors consider this? Given the cell cycle's critical role in DNA repair, understanding whether T4 polymerase affects cell cycle progression is crucial to interpreting the observed editing outcomes.
4. The role of the MS2 coat protein component in the construct should be clarified - its inclusion is noted but then it is never explained or used in the experiments. Did sgRNAs have MS2 stem-loops inserted for polymerase recruitment?
5. Using known guides/targets/translocations is useful in support of the claim that CasPlus suppresses translocations. However, based on the experiments shown that involve the use of *two* locus-specific PCR primers, it is difficult to rule out the formal (if admittedly unlikely) possibility that CasPlus *alters* translocation occurrences without actually suppressing them. In the future the authors should consider using tagmentation-based approaches, which use only a single locus-specific PCR primer, for more

comprehensive and less biased analyses.

Minor issues:

1. As noted by the authors, both Cas9 and T4 DNA polymerase display a preference for 1-bp insertions, with a notable presence of adenine (A) (68.8%) at position -4 within the majority of tested endogenous sites. However, the reporter line used throughout the study (tdTomato-d151A) inadvertently carries a 1-bp deletion of A at position -4. Would it raise the possibility that the reporter line may inadvertently exaggerate insertion efficiency during FACS (tdTomato+) sorting and enrichment, due to mutation bias, whereas endogenous sites deep seq data might compensate for this limitation?
2. "MS2" in protein constructs should be changed to "MCP" in the figures and text.

I am writing to respond to the comments and suggestions provided by the reviewers on our manuscript (EMBOJ-2023-114902R). First and foremost, we would like to express our sincere gratitude to the editor and the reviewers for the effort invested in the thorough evaluation of our work. Reviewers' insightful comments have undeniably enhanced the quality and rigor of our research. We carefully considered each comment and suggestion and have made the necessary revisions to address all the concerns. We outline our responses to the reviewers' comments and detail the changes we made to the manuscript:

Referee #1:

The authors study the use of various DNA polymerases to bias the outcome of CRISPR induced DSB repair towards nucleotide insertions rather than deletions in human cells. They find that the coexpression of Cas9 with T4 or RB69 DNA polymerase leads to strongly increased rates of Insertions and reduced frequencies of deletions and translocations. Using engineered versions of Cas9 and T4 Pol the Insertion frequencies can be further enhanced. This modified gene editing approach named CasPlus by the authors can be of great use for therapeutic reframing of mutant alleles (as demonstrated for DMD mutations) or the therapeutic knockout of modifier genes in CAR T cells while reducing undesired on target effects. Therefore this work will be of interest for the mammalian gene editing community in particular for therapeutic applications. The work will be well received by the audience of a journal that has a focus on gene editing, gene therapy or technical development.

-Thanks for the enthusiastic support!

Referee #2:

In this study, the researchers investigated the influence of different DNA polymerases on Cas9 and Cas12a genome editing outcomes, revealing that T4 DNA polymerase increases the occurrence of small insertions, while simultaneously reducing the frequency of large deletions and chromosomal translocations. Their approach was applied to diverse disease cell models, encompassing pluripotent stem cell-derived cardiomyocytes (DMD), mouse germline editing, and primary human T cells. The study is both elegant and comprehensive, and the data presented are abundant. This will be a useful contribution to the genome editing literature once the following issues are addressed.

We greatly appreciate the positive and constructive feedback provided by the reviewer, which has significantly contributed to the improvement of our manuscript and to guide our future work.

Major points:

1. Supplementary Fig. 1 provides insight into the impact of Cas9 tethering on Pol function. While the authors speculate that the exonuclease domain, rather than the DNA polymerase domain, influences DSB repair and justifies this with strong deletions induced by tethered Cas9 and Pol, evidence supporting this claim remains thin. It would have been very straightforward to completely inactivate the polymerase domain, rather than use only the seven altered-fidelity mutants. This would be an important counterpart to the complete deletion of the exonuclease domain and is crucial to solidify their claim.

-We generated a polymerase domain inactive T4 DNA polymerase (N214S). As shown in appendix Fig. S1I and J, Cas9 fused with T4 Pol (N214S) generate similar strong deletions compared to Cas9 fused with wild-type T4 Pol. Additionally, we tested exonuclease domain inactive T4 DNA polymerase (D219A). Cas9 fused with T4 Pol (D219A) eliminate the increased frequency of deletions induced by Cas9 fused with wild-type T4 Pol. These results support our original hypothesis that the exonuclease domain, rather than the DNA polymerase domain, mainly influences DSB repair and generates strong deletions when T4 DNA polymerase fused to Cas9.

2. Supplementary Fig. 1g,h suggests that the orientation (Cas9-L-T4 vs. L-T4-Cas9) potentially impacts editing outcomes (deletions, etc.). Investigating whether T4 Pol fusion at the N or C terminus influences Cas9 cleavage efficiency, DSB repair, or both, would offer valuable insights. Furthermore, including a T4/Cas9 untethered group for comparison in Supplementary Fig. 1g,h would enhance clarity.

-We added the frequencies of indels induced by Cas9, Cas9-L-T4 or T4-L-Cas9 at tdTomato-d151A site into appendix Fig. S1H. T4 Pol fusion at the N or C terminus strongly favors deletions without substantially influencing Cas9 cleavage efficiency. T4/Cas9 untethered group results were also added in appendix Fig. S1G and H.

3. It would be valuable to explore potential cytotoxic effects of exogenous T4 polymerase and its influence on the cell cycle. Did the authors consider this? Given the cell cycle's critical role in DNA repair, understanding whether T4 polymerase affects cell cycle progression is crucial to interpreting the observed editing outcomes.

-New results were added in appendix Fig. S3I. T4 polymerase treatment does not significantly affect cell cycle progression during the Cas9-mediated genome editing ($p > 0.05$).

4. The role of the MS2 coat protein component in the construct should be clarified - its inclusion is noted but then it is never explained or used in the experiments. Did sgRNAs have MS2 stem-loops inserted for polymerase recruitment?

-New MS2-guide RNA/MCP-T4 Pol results were added in appendix Fig. S1K. Yes, MS2-guide RNA has MS2 loops inserted for MCP-T4 polymerase recruitment. We originally designed this system to facilitate potential in vivo work in the future, since the protein expression levels will be significantly lower in vivo. In our proof-of principle in vitro systems (293T and iPSCs), since the protein expression is saturated, hence it won't be a surprise that there is no significant difference between MS2-guide RNA/MCP-T4 Pol system and guide RNA/MCP-T4 Pol system.

5. Using known guides/targets/translocations is useful in support of the claim that CasPlus suppresses translocations. However, based on the experiments shown that involve the use of *two* locus-specific PCR primers, it is difficult to rule out the formal (if admittedly unlikely) possibility that CasPlus *alters* translocation occurrences without actually suppressing them. In the future the authors should consider using tagmentation-based approaches, which use only a single locus-specific PCR primer, for more comprehensive and less biased analyses.

-We agree that in the future, tagmentation-based approaches will offer us better comprehensive pictures of genome-wide editing. This approach might take a few months for new sample preparation and data analysis; hence it might be beyond the scope of this manuscript. We are more than happy to take this suggestion in our future work.

Minor issues:

1. As noted by the authors, both Cas9 and T4 DNA polymerase display a preference for 1-bp insertions, with a notable presence of adenine (A) (68.8%) at position -4 within the majority of tested endogenous sites. However, the reporter line used throughout the study (tdTomato-d151A) inadvertently carries a 1-bp deletion of A at position -4. Would it raise the possibility that the reporter line may inadvertently exaggerate insertion efficiency during FACS (tdTomato+) sorting and enrichment, due to mutation bias, whereas endogenous sites deep seq data might compensate for this limitation?

-We designed the tdTomato-d151A reporter cell line with the aim to use flow cytometry/deep sequencing assays to screen DNA polymerases that could fill in Cas9-induced 5' overhangs, thus favoring 1-2 bps insertions over deletions. We generated one adenine deletion at position -4 within tdTomato target site since regular Cas9 with guide RNAs containing an A/T at position -4 may likely produce DSBs with 5' overhangs (PMID:30480667 and PMID:30405244). We agreed that FACS sorting, and cell enrichment into two populations may potentially exaggerate insertion efficiency, but the trend of increase in insertion efficiency and decrease in deletions (or vice versa) induced by Cas9 and DNA Pol would be unlikely obviously impacted by sorting and enrichment. To confirm the screen results, we analyzed additional 26 endogenous sites. All data are shown in appendix Fig. S1.

2. "MS2" in protein constructs should be changed to "MCP" in the figures and text.

-Thanks! Done.

Dear Dr. Long,

Thank you for sending us your manuscript revised based on the two reviewer reports I had initially transmitted to you with my original decision of "Revise and Re-review". At that time, I also mentioned a third referee who had been unable to finish their report in time, due to important personal reasons. This referee later still submitted their comments, which admittedly contained a number of well-taken, valid concerns not raised by the other referees. Unfortunately, my subsequent e-mail informing you about these additional comments was not sent to you by our system due to technical problems, something I only realized now when receiving your revision. We very much apologize for this mistake on our part. Nevertheless, I am afraid that addressing some of referee 3's points prior to publication of this study would still appear highly relevant.

This third referee did raise concerns both regarding novelty (Yoo et al 2022 in NAR) as well as several technical aspects. Given that we had already invited revisions before receiving these comments, we shall be happy to overrule the novelty concerns and not take them into account here, however we feel that incorporating the technical points raised by this referee would benefit the manuscript greatly.

In particular, we find that the suggestions regarding further technical validation of the tdTomato reporter assay and the questions regarding the transfection efficiencies of the GFP-tagged constructs in two independent plasmids are both important to the core conclusions of this work.

This referee also suggests further experiments to strengthen the therapeutical aspect of the manuscript. While we think that these would be a great addition, we would not require you addressing these experimentally given the circumstances.

Let me emphasize that we remain committed to proceed with this manuscript, but we think it would be important to address these concerns to make sure that the results reported here stand strong on their own and will be considered a significant contribution to the field. We are aware that this request likely comes at a surprising time and will add additional delay to the publication. Therefore, we will not give you a specific additional deadline here, and we also guarantee that we will not consider any competing studies published or submitted in the meantime for the evaluation of this manuscript.

Should you have any further questions regarding this decision or the additional referee points, please do contact me with and I would be happy to discuss these possible revisions by email or videoconferencing.

Best regards,

Cornelius Schneider

Cornelius Schneider, PhD
Editor | The EMBO Journal
c.schneider@embojournal.org

Report referee #3

General Summary & Significance

Cas9-induced DSBs are repaired by various error-prone DNA repair pathways (eg cNHEJ, altNHEJ, MMEJ). The resultant lesions are somewhat predictable, however the uncontrolled diversity of editing outcomes, including rare larger deletions (and translocations in the context of multiplex editing), limit the clinical application of DSB-inducing CRISPR technologies.

Although the majority of Cas9-cleavage events produce blunt double-strand DNA breaks (DSBs), a subset of Cas9-induced DSBs result in a staggered DSB, and these are handled differently by the DNA repair machinery. A fill-in reaction by endogenous polymerases and components of the NHEJ/MMEJ machinery generates small, templated (1-3 bp) insertions at the repair site with surprising precision. These specific insertions have been demonstrated to be more precise than deletion events and have been used to generate and cure a range of preclinical models of disease. Enhancement of small insertions has been demonstrated through both mutagenesis of key residues in Cas9 (Shou et al 2018 Mol Cell) and through introduction of exogenous DNA polymerases (Woo et al 2022 NAR). An added benefit of biasing repair towards insertions, means that uncontrolled, deleterious large deletions are biased against; and important step towards the clinic.

The authors present a study that largely supports previous findings that the indel spectrum resulting from the repair of Cas9-induced double-strand DNA breaks (DSBs) can be manipulated through a combination of Cas9 mutagenesis and the introduction of exogenous DNA polymerases. The main findings of the study are that exogenously introduced T4 DNA polymerase reduces large deletions and chromosomal translocations through biasing repair towards 1 bp and 2 bp insertions, and that this can be harnessed for therapeutic applications.

The authors follow closely the roadmap laid by Woo et al (2022 NAR) who demonstrated (i) a reduction of large deletions and (ii) an enhancement of 1 bp insertions upon exogenous introduction of Cas9-DNA Pol I or Cas9-Klenow fusions (Klenow being a

fragment of DNA pol I). Further, Woo and colleagues used this approach to demonstrate therapeutic potential in Duchenne muscular dystrophy (DMD).

Specific comments

Suitability of the model system for detecting indel-altering activities:

The authors use an out-of-frame tdTomato cassette as a reporter system to read out the impact on frameshifting due to candidate Polymerase over-expression. This reviewer believes conclusions drawn from this model system are not entirely supported by the data.

1. The data in Figure 1d casts doubt on the fidelity and utility of the tdTomato reporter assay. The +1bp insertion in the Tomato positive pool (40.37%) is also present at high levels in the Tomato negative fraction (12.46%). Further, the predominant indel (-1bp) in the Tomato negative population (32.26%) is the same as the third most prevalent indel in the Tomato positive population (7.83%). It is not clear how the same mutation can lead to both tdTomato positive and negative outcomes. These two data points demonstrate that the reporter assay is flawed. The only reasonable explanation would be that there are multiple integration events of the reporter construct in the cell line - in which case the line ceases to be useful.

2. The reporter assay relies on the introduction of two plasmids, one encoding Cas9 + guide RNA, and the other encoding a DNA Pol variant. Only when both plasmids are in a cell does the impact of the DNA Pol influence the indel spectrum. As noted in Fig 1 and the methods, both plasmids contain a GFP reporter as a marker of transfection. This adds a layer of complexity that limits the interpretability of the assay. As the chance of a plasmid being introduced to a given cell via lipofection is an independent probability, the chance of getting both plasmids in the same cell is comparatively low. Assuming a 50% transfection efficiency, only 25% of the cells would likely contain both plasmids. Enriching for GFP using FACS will only increase this to 50% dual-transfected cells (with the other 50% being singly transfected). Thus, the editing outcomes measured are in fact a blended combination of single and double plasmid transfected cells, resulting in: (i) no editing (DNA Pol plasmid alone), (ii) wt Cas9 indel spectra (pX458 alone) or (ii) DNA Pol influenced indels (dual transfection). Further, as the DNA Pol plasmids differ in size, it is reasonable to assume that they will differ in transfection efficiencies, adding yet more confusion to the result.

Perhaps this offers some insight as to why the authors were not able to reproduce the Cas9-DNA Pol I and Cas9-Klenow results from Woo et al (2022) in Fig 1e. Either a separate colour (eg BFP), or stable lines expressing the polymerases, should be used to add clarity to the impact of each DNA Pol on indel spectra.

The authors appear to imply that the fusion of T4 to Cas9 causes greater frequencies of larger deletions than the MS2-T4. This is in contrast to Woo et al (2022) which showed that fusions of Cas9 to DNA Pol I actually reduced the larger deletion events. Can the authors please show a direct comparison to demonstrate their claim, and discuss the discordance with the 2022 paper. Perhaps the authors have examined the large deletions via NGS of the delta 1-377 variant? This would be interesting to see.

Mechanism:

Why in Fig 1 did T4 enhance exclusively +2 bps, and yet in Sup Fig 2b it had little or no effect on +2 bps? Similarly, in Sup fig 1i T4 exclusively enhanced +1 bps, and in sup fig 1k it was mostly -1 bps, and yet in other sites greater than 1 editing outcome was enhanced (often, but not always, +1 and -1s). This suggests a complexity at play that's not fully described in the paper, nor controlled by the addition of T4 (or mutants thereof). This complexity needs discussed in greater depth to guide the reader.

Reproducibility

A significant omission is the method used to choose the TS's throughout the paper. It is imperative that the gRNA selection process is included, whether that be from a publication or database (with selection methods highlighted) or from a computational tool (with thresholds, metrics and assumptions detailed enough for readers to replicate the work).

Therapeutic application - DMD:

Guide RNA G10 (Fig3) reframes DMD early in Exon 51 meaning that a significant stretch of non-natural amino acids will be incorporated into the final DMD protein. Can the authors please speculate on the impact of these non-wild type amino acids on function (this would be helped by incorporating AlphaFold 2 predictions).

The manuscript focuses on the comparison between an evolved/mutated Cas9 and wild type Cas9 for a large part, particularly when examining therapeutically relevant disease models (DMD, CAR-T). However, this is an artificial comparison as the field favours high-fidelity Cas9 variants for preclinical and clinical therapeutic applications. Perhaps a comment could be added to the discussion relating to the relevance of large deletions to HiFi (and other) evolved Cas9s.

Further, given the therapeutic focus of CasPlus, I would encourage the authors to consider several therapeutic challenges, specifically: (i) the inefficiency of co-delivery of a two component system into muscle tissue in situ, (ii) the ability of CasPlus to function in post-mitotic cells (ie myofibers); (iii) whether adding bacteriophage components will further exacerbate the well-described immunogenicity of delivering Cas9 to DMD patients, and (iv) the applicable number of patients that a therapy such as this would realistically reframe (keeping in mind that it will also knock a wild-type allele out of frame!).

Minor comments

Fig 1d - Representative FACS plots should be shown to demonstrate the relative proportions of Tomato+/GFP+ and Tomato-/GFP+ for clarity (please also include FACS plots to demonstrate the leakiness of the reporter system).

p7: a 1-fold change is not an increase

Please make clear in Fig 3 that iCMs have not been edited. Rather they are edited at the iPSC stage prior to differentiation.

P11 - please state the method and reference used to determine predicted off target sites

P12 - for the large deletion analysis, how have you corrected for larger deletions amplifying more than wild type sequence, so as not to over-estimate the PacBio data?

Fig5d is confusing. It seems 10 embryos per group were analysed, however data is shown for 18 or 13 embryos per group.

Fig5 - please include sequencing data and a summary plot of the editing efficiencies for the embryo work, as one explanation could be that there was overall less editing with -V1 and V2.

P17 - while it's true there was a hold put on the CAR-T therapy, in the interests of balance it should be noted that the trial wasn't abandoned. Note also that the FDA ruled that junction PCRs (as used in this paper) were not sufficient to detect chromosomal translocations and preferred the use of unbiased methods (eg genome-wide optical FISH technologies)

It seems improbable that the best two gRNAs computationally predicted to yield robust 1 bp insertions would result in only 5% and ~18% +1 bp using Cas9. Please show the data from all gRNAs tested across Exon's 51 and 53 with Cas9.

It is clear from Fig 6 that translocations are still a problem for CasPlus. One solution to this might be to deliver the edits sequentially rather than in parallel.

Please remember: Digital image enhancement is acceptable practice, as long as it accurately represents the original data and conforms to community standards. If a figure has been subjected to significant electronic manipulation, this must be noted in the

figure legend or in the 'Materials and Methods' section. The editors reserve the right to request original versions of figures and the original images that were used to assemble the figure.

We realize that it is difficult to revise to a specific deadline. In the interest of protecting the conceptual advance provided by the work, we recommend a revision within 3 months (7th Feb 2024). Please discuss the revision progress ahead of this time with the editor if you require more time to complete the revisions. Use the link below to submit your revision:

I am writing to respond to the comments and suggestions provided by the reviewers on our manuscript. First and foremost, we would like to express our sincere gratitude to the editor and the reviewers for the effort invested in the thorough evaluation of our work. Reviewers' insightful comments have undeniably enhanced the quality and rigor of our research. We carefully considered each comment and suggestion and have made the necessary revisions to address the concerns. We outline our responses to the reviewers' comments and detail the changes we made to the manuscript:

Referee #1:

The authors study the use of various DNA polymerases to bias the outcome of CRISPR induced DSB repair towards nucleotide insertions rather than deletions in human cells. They find that the coexpression of Cas9 with T4 or RB69 DNA polymerase leads to strongly increased rates of Insertions and reduced frequencies of deletions and translocations. Using engineered versions of Cas9 and T4 Pol the Insertion frequencies can be further enhanced. This modified gene editing approach named CasPlus by the authors can be of great use for therapeutic reframing of mutant alleles (as demonstrated for DMD mutations) or the therapeutic knockout of modifier genes in CAR T cells while reducing undesired on target effects. Therefore this work will be of interest for the mammalian gene editing community in particular for therapeutic applications. The work will be well received by the audience of a journal that has a focus on gene editing, gene therapy or technical development.

-Thanks for the enthusiastic support!

Referee #2:

In this study, the researchers investigated the influence of different DNA polymerases on Cas9 and Cas12a genome editing outcomes, revealing that T4 DNA polymerase increases the occurrence of small insertions, while simultaneously reducing the frequency of large deletions and chromosomal translocations. Their approach was applied to diverse disease cell models, encompassing pluripotent stem cell-derived cardiomyocytes (DMD), mouse germline editing, and primary human T cells. The study is both elegant and comprehensive, and the data presented are abundant. This will be a useful contribution to the genome editing literature once the following issues are addressed.

We greatly appreciate the positive and constructive feedback provided by the reviewer, which has significantly contributed to the improvement of our manuscript.

Major points:

1. Supplementary Fig. 1 provides insight into the impact of Cas9 tethering on Pol function. While the authors speculate that the exonuclease domain, rather than the DNA polymerase domain, influences DSB repair and justifies this with strong deletions induced by tethered Cas9 and Pol, evidence supporting this claim remains thin. It would have been very straightforward to completely inactivate the polymerase domain, rather than use only the seven altered-fidelity mutants. This would be an important counterpart to the complete deletion of the exonuclease domain and is crucial to solidify their claim.

-We generated polymerase domain inactive T4 DNA polymerase (N214S). As shown in appendix **new panels Fig. S1J and K**, Cas9 fused with T4 Pol (N214S) generate similar strong deletions compared to Cas9 fused with wild-type T4 Pol. Additionally, we tested exonuclease domain inactive T4 DNA polymerase (D219A). Cas9 fused with T4 Pol (D219A) eliminate the increased frequency of deletions induced by Cas9 fused with wild-type T4 Pol. These results support our original hypothesis that the exonuclease domain, rather than the DNA polymerase domain, mainly influences DSB repair and generates strong deletions when T4 DNA polymerase fused to Cas9.

2. Supplementary Fig. 1g,h suggests that the orientation (Cas9-L-T4 vs. L-T4-Cas9) potentially impacts editing outcomes (deletions, etc.). Investigating whether T4 Pol fusion at the N or C terminus influences Cas9 cleavage efficiency, DSB repair, or both, would offer valuable insights. Furthermore, including a T4/Cas9 untethered group for comparison in Supplementary Fig. 1g,h would enhance clarity.

-We added the frequencies of indels induced by Cas9, Cas9-L-T4 or T4-L-Cas9 at tdTomato-d151A site into appendix **new panel Fig. S1I**. T4 Pol fusion at the N or C terminus strongly favors deletions without substantially influencing Cas9 cleavage efficiency. T4/Cas9 untethered group results were also added in appendix **Fig. S1H and I**.

3. It would be valuable to explore potential cytotoxic effects of exogenous T4 polymerase and its influence on the cell cycle. Did the authors consider this? Given the cell cycle's critical role in DNA repair, understanding whether T4 polymerase affects cell cycle progression is crucial to interpreting the observed editing outcomes.

-New results were added in appendix **new panel Fig. S3I**. T4 polymerase does not significantly affect cell cycle progression during the Cas9-mediated genome editing ($p > 0.05$).

4. The role of the MS2 coat protein component in the construct should be clarified - its inclusion is noted but then it is never explained or used in the experiments. Did sgRNAs have MS2 stem-loops inserted for polymerase recruitment?

-New MS2-guide RNA/MCP-T4 Pol results were added in appendix **new panel Fig. S1L**. Yes, MS2-guide RNA has MS2 loops inserted for MCP-T4 polymerase recruitment. We originally designed this system to facilitate potential in vivo work since the protein expression levels will be significantly lower in tissue. In our proof-of principle in vitro systems (293T and iPSCs), since the protein expression is saturated, hence it won't be a surprise that there is no significant difference between MS2-guide RNA/MCP-T4 Pol system and guide RNA/MCP-T4 Pol system.

5. Using known guides/targets/translocations is useful in support of the claim that CasPlus suppresses translocations. However, based on the experiments shown that involve the use of *two* locus-specific PCR primers, it is difficult to rule out the formal (if admittedly unlikely) possibility that CasPlus *alters* translocation occurrences without actually suppressing them. In the future the authors should consider using tagmentation-based approaches, which use only a single locus-specific PCR primer, for more comprehensive and less biased analyses.

-We agree that in the future, tagmentation-based approaches will offer us better comprehensive pictures of genome-wide editing. This approach might take a few months for new sample preparation and data analysis; hence it might be beyond the scope of this manuscript. We are more than happy to take this suggestion in our future work.

Minor issues:

1. As noted by the authors, both Cas9 and T4 DNA polymerase display a preference for 1-bp insertions, with a notable presence of adenine (A) (68.8%) at position -4 within the majority of tested endogenous sites. However, the reporter line used throughout the study (tdTomato-d151A) inadvertently carries a 1-bp deletion of A at position -4. Would it raise the possibility that the reporter line may inadvertently exaggerate insertion efficiency during FACS (tdTomato+) sorting and enrichment, due to mutation bias, whereas endogenous sites deep seq data might compensate for this limitation?

- We designed this reporter cell line with the aim to use flow cytometry/deep sequencing assays to screen DNA polymerases that could fill in Cas9-induced 5' overhangs, thus favoring 1-2 bps insertions over deletions. We deleted one A at position -4 within tdTomato gene because regular Cas9 with guide RNAs containing an A/T at position -4 may likely produce DSBs with 5' overhangs (PMID: 30480667 and PMID: 30405244). We agree that FACS sorting, and cell enrichment into two populations may potentially exaggerate insertion efficiency, but the trend of increase in insertion efficiency and decrease in deletions (or vice versa) induced by Cas9 and DNA Pol would be unlikely obviously impacted by sorting and enrichment. To confirm the screen results, we analyzed 26 endogenous sites. All data are shown in **appendix Fig. S1**.

2. "MS2" in protein constructs should be changed to "MCP" in the figures and text.

-Done

referee #3

General Summary & Significance

Cas9-induced DSBs are repaired by various error-prone DNA repair pathways (eg cNHEJ, altNHEJ, MMEJ). The resultant lesions are somewhat predictable, however the uncontrolled diversity of editing outcomes, including rare larger deletions (and translocations in the context of multiplex editing), limit the clinical application of DSB-inducing CRISPR technologies.

Although the majority of Cas9-cleavage events produce blunt double-strand DNA breaks (DSBs), a subset of Cas9-induced DSBs result in a staggered DSB, and these are handled differently by the DNA repair machinery. A fill-in reaction by endogenous polymerases and components of the NHEJ/MMEJ machinery generates small, templated (1-3 bp) insertions at the repair site with surprising precision. These specific insertions have been demonstrated to be more precise than deletion events and have been used to generate and cure a range of preclinical models of disease. Enhancement of small insertions has been demonstrated through both mutagenesis of key residues in Cas9 (Shou et al 2018 Mol Cell) and through introduction of exogenous DNA polymerases (Woo et al 2022 NAR). An added benefit of biasing repair towards insertions, means that uncontrolled, deleterious large deletions are biased against; and important step towards the clinic.

The authors present a study that largely supports previous findings that the indel spectrum resulting from the repair of Cas9-induced double-strand DNA breaks (DSBs) can be manipulated through a combination of Cas9 mutagenesis and the introduction of exogenous DNA polymerases. The main findings of the study are that exogenously introduced T4 DNA polymerase reduces large deletions and chromosomal translocations through biasing repair towards 1 bp and 2 bp insertions, and that this can be harnessed for therapeutic applications.

The authors follow closely the roadmap laid by Woo et al (2022 NAR) who demonstrated (i) a reduction of large deletions and (ii) an enhancement of 1 bp insertions upon exogenous introduction of Cas9-DNA Pol I or Cas9-Klenow fusions (Klenow being a fragment of DNA pol I). Further, Woo and colleagues used this approach to demonstrate therapeutic potential in Duchenne muscular dystrophy (DMD).

-Thanks for highlighting the significance and pointing the future directions of our CasPlus projects.

Specific comments

Suitability of the model system for detecting indel-altering activities:

The authors use an out-of-frame tdTomato cassette as a reporter system to read out the impact on frameshifting due to candidate Polymerase over-expression. This reviewer believes conclusions drawn from this model system are not entirely supported by the data.

1. The data in Figure 1d casts doubt on the fidelity and utility of the tdTomato reporter assay. The +1bp insertion in the Tomato positive pool (40.37%) is also present at high levels in the Tomato negative fraction (12.46%). Further, the predominant indel (-1bp) in the Tomato negative

population (32.26%) is the same as the third most prevalent indel in the Tomato positive population (7.83%). It is not clear how the same mutation can lead to both tdTomato positive and negative outcomes. These two data points demonstrate that the reporter assay is flawed. The only reasonable explanation would be that there are multiple integration events of the reporter construct in the cell line - in which case the line ceases to be useful.

- It is well known that stable cell lines generated via lentivirus can result in integration of a variable numbers of copies of targeted DNA into the genome. We totally agree with this reviewer's explanation that tdTomato reporter cell line might contain multiple copies of reporter gene. We added this explanation to our main text on **Page 6**.

Although we did not measure how many functional copies of tdTomato-d151A are in the genome, we assumed that the Tomato-positive pool will ENRICH cells with re-framed tdTomato genes ($3n+1$), while the Tomato-negative pool will ENRICH out-of-frame tdTomato genes ($3n$ and $3n+2$). Through initial screening using the same cell line, we found that T4 DNA polymerase overwhelmingly outperformed other DNA polymerases in favoring insertions over deletions. An additional 26 endogenous genes' editing results further confirmed the reliability of using this reporter for initial screening.

2. The reporter assay relies on the introduction of two plasmids, one encoding Cas9 + guide RNA, and the other encoding a DNA Pol variant. Only when both plasmids are in a cell does the impact of the DNA Pol influence the indel spectrum. As noted in Fig 1 and the methods, both plasmids contain a GFP reporter as a marker of transfection. This adds a layer of complexity that limits the interpretability of the assay. As the chance of a plasmid being introduced to a given cell via lipofection is an independent probability, the chance of getting both plasmids in the same cell is comparatively low. Assuming a 50% transfection efficiency, only 25% of the cells would likely contain both plasmids. Enriching for GFP using FACS will only increase this to 50% dual-transfected cells (with the other 50% being singly transfected). Thus, the editing outcomes measured are in fact a blended combination of single and double plasmid transfected cells, resulting in: (i) no editing (DNA Pol plasmid alone), (ii) wt Cas9 indel spectra (pX458 alone) or (iii) DNA Pol influenced indels (dual transfection). Further, as the DNA Pol plasmids differ in size, it is reasonable to assume that they will differ in transfection efficiencies, adding yet more confusion to the result.

-Thanks for pointing this out. We understand that GFP enrichment assay of co-transfection experiments could produce the above three possibilities, potentially resulting in a reduction of entire editing efficiency. To ensure the different editing profiles between T4 DNA polymerase and other DNA polymerase was not mainly due to variable transfection efficiency and protein expression, we performed Western blot and verified the expression levels of all the DNA polymerase tested (**Appendix Fig S1F**).

With this carefully controlled experiment, we observed a 6-fold increase of frequency in 2-bp insertions only in T4 DNA polymerase and Cas9 co-expressed versus Cas9-only tdTomato-GFP+ cells, revealing the strong effect of T4 Pol on Cas9 editing outcomes (**Fig 1E**).

Perhaps this offers some insight as to why the authors were not able to reproduce the Cas9-DNA Pol I and Cas9-Klenow results from Woo et al (2022) in Fig 1e. Either a separate colour

(eg BFP), or stable lines expressing the polymerases, should be used to add clarity to the impact of each DNA Pol on indel spectra.

-Yoo et al used fusing Escherichia coli DNA polymerase I or the Klenow fragment to Cas9 while we are using untethered DNA polymerases and Cas9 group hence editing profiles might be different.

It worth to noting that fused T4 DNA polymerase-Cas9 generated relatively larger deletions (**Appendix Fig S1G-K**) while Yoo et al observed fusing DNA polymerase I or the Klenow fragment to Cas9 significantly increases the frequencies of **1-bp deletions and decreases >1-bp deletions or insertions**. These results further underscored the difference between our CasPlus system to Yoo's group.

The authors appear to imply that the fusion of T4 to Cas9 causes greater frequencies of larger deletions than the MS2-T4. This is in contrast to Woo et al (2022) which showed that fusions of Cas9 to DNA Pol I actually reduced the larger deletion events. Can the authors please show a direct comparison to demonstrate their claim and discuss the discordance with the 2022 paper. Perhaps the authors have examined the large deletions via NGS of the delta 1-377 variant? This would be interesting to see.

-Comprehensively and systematically comparing editing profiles of different DNA polymerases fusing to Cas9 is a very interesting topic! As explained above, our manuscript's primary goal is to develop T4 DNA polymerase as a new tool for gene editing, hence these comparisons might be beyond the scope of this paper. We discussed the difference between ours and Yoo et al's in the discussion part (**Page 18**).

Mechanism:

Why in Fig 1 did T4 enhance exclusively +2 bps, and yet in Sup Fig 2b it had little or no effect on +2 bps? Similarly, in Sup fig 1i T4 exclusively enhanced +1 bps, and in sup fig 1k it was mostly -1 bps, and yet in other sites greater than 1 editing outcome was enhanced (often, but not always, +1 and -1s). This suggests a complexity at play that's not fully described in the paper, nor controlled by the addition of T4 (or mutants thereof). This complexity needs discussed in greater depth to guide the reader.

- It is well known that the editing outcome of Cas9 editing is guide RNA/targeted sequence context-dependent, as is CasPlus editing. The majority of double-strand breaks (DSBs) created by Cas9 result in blunt ends, with a subset of them having staggered ends with 5' overhangs. For guide RNAs capable of inducing 5' overhangs of 1-2 bps, T4 DNA polymerase could directly fill in the overhangs, resulting in 1-2 bps insertions. In the case of guide RNAs producing blunt ends, T4 DNA polymerase favors 1-2 bp small deletions, potentially counteracting other DNA repair processes such as end resection. The targeted sequences of the guide RNAs used in Fig 1 or Fig 2 are different, leading to distinct outcomes. In **Appendix Fig S1N-R**, we categorized the tested guide RNAs into different groups based on the editing outcomes. We further discussed the complexity of editing outcomes in the Discussion section (**Page 18**).

Reproducibility

A significant omission is the method used to choose the TS's throughout the paper. It is imperative that the gRNA selection process is included, whether that be from a publication or database (with selection methods highlighted) or from a computational tool (with thresholds, metrics and assumptions detailed enough for readers to replicate the work).

-Thanks. We modified the description on **Page 8**. We chose these guides targeting multiple pathogenic genes which are studied in our lab.

Therapeutic application - DMD:

Guide RNA G10 (Fig3) reframes DMD early in Exon 51 meaning that a significant stretch of non-natural amino acids will be incorporated into the final DMD protein. Can the authors please speculate on the impact of these non-wild type amino acids on function (this would be helped by incorporating AlphaFold 2 predictions).

- DMD stands out as one of the therapeutic applications of CasPlus editing. In Fig 3, we present proof-of-concept evidence demonstrating that multiple guide RNAs can be employed for efficient CasPlus editing, correcting DMD with exon 52 deletions in vitro. To determine the optimal guide RNA, more comprehensive in vivo functional analysis and/or bioinformatics predictions are necessary. However, such an undertaking might go beyond the scope of this paper.

The manuscript focuses on the comparison between an evolved/mutated Cas9 and wild type Cas9 for a large part, particularly when examining therapeutically relevant disease models (DMD, CAR-T). However, this is an artificial comparison as the field favours high-fidelity Cas9 variants for preclinical and clinical therapeutic applications. Perhaps a comment could be added to the discussion relating to the relevance of large deletions to HiFi (and other) evolved Cas9s.

-Very good suggestions! Our proof-of-concept study primarily focused on assessing the impact of T4 DNA polymerase on **on-target** Cas9 editing, utilizing the widely used wild-type Cas9 for a significant portion of the investigation. High-fidelity Cas9 variants are primarily designed to reduce **off-target** effects. Additionally, we explored the capacity of T4 DNA polymerase to favor insertions over deletions in other mutated Cas9 variants and other types of Cas proteins, such as Cas12a, which can produce short 5' staggered ends. Since T4 DNA polymerase demonstrated a preference for insertions over deletions when combined with all tested Cas9 variants or Cas proteins, we speculate that similar results can be achieved when combined with high-fidelity (HiFi) or other evolved Cas9 variants, provided these variants create staggered ends at the target sites. A comment addressing this speculation has been added on **Page 18**.

Further, given the therapeutic focus of CasPlus, I would encourage the authors to consider several therapeutic challenges, specifically: (i) the inefficiency of co-delivery of a two component system into muscle tissue in situ, (ii) the ability of CasPlus to function in post-mitotic cells (ie myofibers); (iii) whether adding bacteriophage components will further exacerbate the well-described immunogenicity of delivering Cas9 to DMD patients, and (iv) the applicable number of

patients that a therapy such as this would realistically reframe (keeping in mind that it will also knock a wild-type allele out of frame!).

-Thanks for the great suggestions. We will take these challenges into consideration when performing subsequent in vivo DMD studies.

Minor comments

Fig 1d - Representative FACS plots should be shown to demonstrate the relative proportions of Tomato+/GFP+ and Tomato-/GFP+ for clarity (please also include FACS plots to demonstrate the leakiness of the reporter system).

-Representative FACS plots were added in appendix **new panel Fig S1A**.

p7: a 1-fold change is not an increase.

-Thanks for pointing this out. We modified the description of the sentence on **Page 9**.

Please make clear in Fig 3 that iCMs have not been edited. Rather they are edited at the iPSC stage prior to differentiation.

-We modified the figure legends for **Fig 3** to clarify that editing was performed in the iPSCs stage.

P11 - please state the method and reference used to determine predicted off target sites.

-The method to predict off-target sites and to analyze the genome-wide off-target were shown in Material and Methods. The reference was cited in the main text (**Page 13**) and the Material and Methods.

P12 - for the large deletion analysis, how have you corrected for larger deletions amplifying more than wild type sequence, so as not to over-estimate the PacBio data?

-We acknowledge that biased PCR amplifications of DNA alleles with smaller sizes than larger sizes pose a challenge for PacBio sequencing, as well as other sequencing-based technologies. In this study, our aim was to compare the trends in the frequency of large deletions induced by CasPlus versus Cas9. In Fig 4, our PacBio data revealed that CasPlus resulted in a relatively decreased frequency of large deletions compared to Cas9. To precisely quantify the absolute efficiency of large deletions induced by Cas9 or CasPlus, more comprehensive and unbiased methods can be applied in future studies.

Fig5d is confusing. It seems 10 embryos per group were analyzed, however data is shown for 18 or 13 embryos per group.

-We totally tested 10 embryos per group. All the 10 embryos with Cas9 or CasPlus editing contained deletions with the size <500 bp, but 8 of 10 embryos with Cas9 editing and 3 of 10 embryos with CasPlus editing carried deletions with the size >500 bp.

Fig5 - please include sequencing data and a summary plot of the editing efficiencies for the embryo work, as one explanation could be that there was overall less editing with -V1 and V2.

-The editing efficiencies of embryos with Cas9 or CasPlus editing were added in appendix **new panel Fig S5A**. The overall editing efficiency of CasPlus is comparable with Cas9. The original sequencing data of Fig 5C were added in appendix **new panel Fig S5C-D**.

P17 - while it's true there was a hold put on the CAR-T therapy, in the interests of balance it should be noted that the trial wasn't abandoned. Note also that the FDA ruled that junction PCRs (as used in this paper) were not sufficient to detect chromosomal translocations and preferred the use of unbiased methods (eg genome-wide optical FISH technologies)

-We modified the description of this citation on **Page 19**. To comprehensively evaluate the genome-wide chromosomal translocations in CAR-T cell generation, we agree that just junction PCRs are not sufficient and unbiased methods are required in the future.

It seems improbable that the best two gRNAs computationally predicted to yield robust 1 bp insertions would result in only 5% and ~18% +1 bp using Cas9. Please show the data from all gRNAs tested across Exon's 51 and 53 with Cas9.

- We generated all gRNAs targeting exon 51 or exon 53, which have the potential to reframe DMD by deleting exon 52, using regular Cas9. Subsequently, we determined their editing efficiencies in HEK293T cells. Next, we conducted CasPlus editing experiments with guide RNAs G10, Ex51-G2, G9, and Ex53-G3, as these induced relatively higher editing efficiencies compared to others in HEK293T cells (**Fig 3C, Appendix Fig S1N-O, and Fig S3A**). To further validate these results, we tested gRNAs G10 and G9 in iPSCs. The original +1 bp frequencies observed in all gRNAs tested across exons 51 and 53 are presented in the updated **Fig 3C**.

It is clear from Fig 6 that translocations are still a problem for CasPlus. One solution to this might be to deliver the edits sequentially rather than in parallel.

- We appreciate the suggestion, and we are more than happy to explore the changes in translocation frequency through the sequential delivery of components in future ex vivo T-cell experiments for CasPlus editing.

Dear Dr. Long,

Thank you for submitting your manuscript for consideration by the EMBO Journal. It has now been seen by referees #2 and #3 again whose comments are shown below. I apologize for the delay in my decision which was caused both by delays in the referee reports as well as by extensive discussion within the team.

As you can see the referees again express in principle interest in the manuscript but do not think that important technical comments raised by referee #3 have been sufficiently addressed and would require additional experiments.

In particular, both referees are not convinced that reliable conclusions can be drawn from the dual reporter system where both factors are tagged with the same marker and in my opinion explicitly ask for additional experiments where cells are sorted for the presence of both Cas9 and T4 Pol.

In addition, referee #3 asks for additional mechanistic insight and further clarification of the observed effects for delivering both factors on different vectors. This is of particular importance as this is a critical finding in the manuscript which differentiates this study from the previous publication by Yoo et al and both referees agree that this would be of strong interest to the field.

We have now re-discussed the manuscript, the revisions, and the referee comments within the team. We generally do not allow for additional rounds of major revisions at the EMBO Journal. Given the unusual circumstances of the delayed referee report we have decided nonetheless to invite you to address the above-mentioned concerns in a second round of revision experiments. We would like to emphasize that the concerns raised by the referees would have to be addressed thoroughly for publication at the EMBO Journal. I am happy to discuss any further questions via email or videoconferencing.

Thank you for the opportunity to consider your work for publication. I look forward to your revision.

Yours sincerely,

Cornelius Schneider, PhD
Editor
The EMBO Journal
c.schneider@embojournal.org

- a point-by-point response to the referees' comments, with a detailed description of the changes made (as a word file).
- a word file of the manuscript text.
- individual production quality figure files (one file per figure)
- a complete author checklist, which you can download from our author guidelines

(<https://www.embopress.org/page/journal/14602075/authorguide>).
- Expanded View files (replacing Supplementary Information)
Please see out instructions to authors
<https://www.embopress.org/page/journal/14602075/authorguide#expandedview>

We realize that it is difficult to revise to a specific deadline. In the interest of protecting the conceptual advance provided by the work, we recommend a revision within 3 months (23rd Apr 2024). Please discuss the revision progress ahead of this time with the editor if you require more time to complete the revisions. Use the link below to submit your revision:

Referee #2:

The concerns that raised in my initial review have been adequately addressed. Another reviewer (#3), however, identified additional issues that are dealt with partially but not completely in the revision.

There are indeed similarities between the current work and Yoo et al. (NAR 2022), but the two works are certainly not identical - the current work has a much heavier focus on a distinct DNA pol (T4) with stronger 3' 5' exonuclease activity. The current paper also applies a broader range of disease cell models (pluripotent stem cell-derived cardiomyocytes, mouse germline editing, and primary human T cells) as compared to Yoo et al (2022).

On the other hand, there are several instances where Reviewer #3 raises substantial technical issues (the use of a multicopy reporter, the lack of two separately discernable fluorescent protein markers in co-transfection experiments, the effects of DNA pol fusion on large deletion frequencies, others) that the authors more-or-less acknowledge in their response but do not effectively answer or, in most instances, address experimentally. This diminishes enthusiasm for the revised work.

Referee #3:

The revised manuscript has made commendable strides in addressing several concerns from the initial review. However, there remain unresolved issues that could potentially lead to misinterpretation of the data by readers.

The modifications to the manuscript have satisfactorily covered the limitations of the reporter assay, specifically in relation to having more than one reporter construct per cell. It is worth noting that while the assay may 'enrich' for 3n+1 events, the same cannot be said for the tdTomato negative fraction.

A significant point of confusion that has not been adequately clarified is the consequence of having both the Cas9 and the candidate polymerases expressed from plasmids with the same (GFP) marker. The observed indel spectra will be an admixture from those derived from Cas9-only containing cells, along with Cas9+pol containing cells. The western blot data does not add any granularity to this confounder. Indeed, the authors have now included indel spectra into Appendix Sup Fig 1H (Tomato negative fraction) that supports the above conjecture, showing that the indel spectra generated from Cas9-pol fusion genes (ie instances where only one plasmid is transfected and each cell will receive both Cas9+pol) show similar indel profiles to each other, and these are distinct from indel profiles derived from Cas9-only and the Cas9 + Pol dual transfections (which both show Cas9-type indel profiles). This confirms to this reviewer that the two-part transfection system (with each plasmid having the same marker) is difficult to interpret. This must be clarified in the text, or preferably an additional colour introduced into the system (eg BFP) to bring a solidity to the data and interpretability of the assay and, in particular, Figure 1.

The inclusion of more expansive data in Appendix Sup Fig 1H-K make clear the authors point that the fusion of Cas9 to T4 pol results in larger deletion events. This is at odds with Yoo et al.

The authors argue that the presence of larger deletions from fusion events necessitates the move towards a two-part MS2 system where T4 pol is delivered in trans. However, in the newly added data in Appendix Sup Fig 1L the authors show that there is no difference between whether a gRNA has an MS2 domain or not, and that simply the presence of T4 pol alone is enough to

stimulate +2 bp insertions. This is contrary to the narrative of the study where it is implied that the MS2::MCP interaction is needed to bring T4 pol into the proximity of the staggered DSB to resolve it and impart its impact on indel spectra.

Given Yoo et al clearly demonstrate the use of a DNA polymerase to counter large deletions and drive an increase in +1 bp insertions (and occasionally +2 bp insertions), to differentiate this study a more precise depiction of the underlying mechanisms is essential. The current manuscript provides minimal mechanistic insights, leaning towards speculation rather than empirical evidence. Data that delineate the mechanistic differences from the Yoo study would significantly bolster the novelty and scientific contribution of this paper.

In summary, while the authors have made efforts to address previous concerns, the presentation and interpretation of the data require further clarity and distinction from prior studies to solidify the manuscript's contribution to the field.

Dear Dr. Schneider,

I am writing to submit the revised version of our manuscript (EMBOJ-2023-114902R1) following the valuable guidance provided by you and the insightful feedback from the reviewers. We are grateful for the thorough review process, which has significantly enhanced the quality and scope of our work.

In response to the key technical comments raised during the review, we have made substantial revisions to the manuscript, focusing on the following aspects:

1.New Dual-fluorescent Reporter System:

We have implemented a new dual-fluorescent reporter system, tagging Cas9 and DNA Polymerase with distinguished markers (BFP and GFP), as illustrated in the newly added **Figure 1 D-G and appendix Fig S1A-D**. New main text to explain this new system was added **Page 6 -8**.

2.Additional Mechanistic Insight and Clarification:

To address the reviewers' queries and provide a more comprehensive understanding of our findings, we have incorporated additional experimental data and an in-depth discussion. This includes the introduction of new appendix panels in appendix fig 1 (**appendix Fig S1O-Q and Fig S1T**) for presenting further mechanistic insights.

Additionally, we have clarified the observed effects when delivering Cas9 and T4 Pol on three different systems, elaborating on this aspect on **Page 6-10**.

We believe that these revisions significantly strengthen the manuscript, addressing the concerns raised during the review process. We hope that these enhancements align with the standards of EMBO J. and contribute positively to the readers.

Referee #2:

The concerns that raised in my initial review have been adequately addressed. Another reviewer (#3), however, identified additional issues that are dealt with partially but not completely in the revision.

-We are delighted to know that all the initial comments from referee #2 are addressed. Now, we are comprehensively addressing referee #3 additional points.

There are indeed similarities between the current work and Yoo et al. (NAR 2022), but the two works are certainly not identical - the current work has a much heavier

focus on a distinct DNA pol (T4) with stronger 3'→5' exonuclease activity. The current paper also applies a broader range of disease cell models (pluripotent stem cell-derived cardiomyocytes, mouse germline editing, and primary human T cells) as compared to Yoo et al (2022).

-We are greatly appreciated that this referee agreed the major focus and scope of our CasPlus work is significantly distinguished from Yoo *et al* work.

As our title indicates: we mainly study Phage DNA polymerases with distinguish enzyme activities from bacteria DNA polymerases. More importantly, we are advancing CasPlus system to multiple systems: human iPSC/cardiomyocytes, mouse germline, human T-cells in multiple diseases models.

On the other hand, there are several instances where Reviewer #3 raises substantial technical issues (the use of a multicopy reporter, the lack of two separately discernable fluorescent protein markers in co-transfection experiments, the effects of DNA pol fusion on large deletion frequencies, others) that the authors more-or-less acknowledge in their response but do not effectively answer or, in most instances, address experimentally. This diminishes enthusiasm for the revised work.

-Thanks for raising these concerns. We are addressing these technical issues with additional experiments and new data which are detailed in the response to referee #3. Here is a quick summary:

-the use of a multicopy reporter

We added explanation in the main text and discussion which satisfied referee #3.

-the lack of two separately discernable fluorescent protein markers in co-transfection experiments

We added the tagging of Cas9 and T4 Pol with distinguished markers (BFP and GFP), as illustrated in the newly added **Figure 1 D-G** and **appendix Fig S1A-D**. Overall, the major conclusion and results are consistent with the original single marker (GFP only) system (**Appendix fig S1E-H**) that T4 and Rb69 DNA polymerases significantly shift the indel profile.

-the effects of DNA pol fusion on large deletion frequencies

New T4 pol and Cas9 fusion data on CLCN5 site, which was used in Yoo et al paper, was added in **appendix fig S1O-P** and **fig S1T**. Our new results are

consistent with the previous conclusion that the fusion of T4 DNA pol and Cas9 protein favors PAM distal deletions.

Referee #3:

The revised manuscript has made commendable strides in addressing several concerns from the initial review. However, there remain unresolved issues that could potentially lead to misinterpretation of the data by readers.

-Many thanks for your recognition of the improvement in our revision. We are more than happy to future clarify these issues to the readers with additional experiments which was suggested by this referee.

The modifications to the manuscript have satisfactorily covered the limitations of the reporter assay, specifically in relation to having more than one reporter construct per cell. It is worth noting that while the assay may 'enrich' for $3n+1$ events, the same cannot be said for the tdTomato negative fraction.

A significant point of confusion that has not been adequately clarified is the consequence of having both the Cas9 and the candidate polymerases expressed from plasmids with the same (GFP) marker. The observed indel spectra will be an admixture from those derived from Cas9-only containing cells, along with Cas9+pol containing cells. The western blot data does not add any granularity to this confounder. Indeed, the authors have now included indel spectra into Appendix Sup Fig 1H (Tomato negative fraction) that supports the above conjecture, showing that the indel spectra generated from Cas9-pol fusion genes (ie instances where only one plasmid is transfected and each cell will receive both Cas9+pol) show similar indel profiles to each other, and these are distinct from indel profiles derived from Cas9-only and the Cas9 + Pol dual transfections (which both show Cas9-type indel profiles). This confirms to this reviewer that the two-part transfection system (with each plasmid having the same marker) is difficult to interpret. This must be clarified in the text, or preferably an additional colour introduced into the system (eg BFP) to bring a solidity to the data and interpretability of the assay and, in particular, Figure 1.

-We greatly appreciated the feedback in our reporter cells. We generated new Fig. 1 with dual fluorescence markers (Cas9-BFP and DNA polymerase-GFP) and conducted a comprehensive comparison of the alterations in indel profiles within BFP⁺GFP⁺ population. Hence, the observed indel spectra will be derived from Cas9 and polymerase containing cells (New panels in **Fig 1D-G**).

Sequencing of the BFP+GFP+ populations revealed that co-expression of T4 or RB69 DNA polymerase with Cas9 resulted in approximately 40% 2-bp insertions, co-

expression of PolII or KF led to approximately 6%-11% 2-bp insertions whereas co-expression of other DNA polymerases with Cas9 or Cas9-only control induced about 2%-4% 2-bp insertions (New panels in **Fig 1D-1F**).

We also analyzed the BFP+GFP+tdTomato+ and BFP+GFP+tdTomato- populations and observed similar trends of alterations in indel profiles alternations (New panels in **appendix Fig S1A-1D**). Overall, when comparing the alternations in indel profiles among BFP+GFP+, BFP+GFP+tdTomato+, and BFP+GFP+tdTomato- populations isolated from experiments using dual fluorescence markers to those in the GFP+tdTomato+ and GFP+tdTomato- populations isolated from earlier experiments utilizing one single fluorescence marker (**Appendix Fig S1E-1H**), we obtained consistent results. These findings collectively suggest that T4 DNA polymerase favors small insertions over deletions on this target site.

The inclusion of more expansive data in Appendix Sup Fig 1H-K make clear the authors point that the fusion of Cas9 to T4 pol results in larger deletion events. This is at odds with Yoo et al.

The authors argue that the presence of larger deletions from fusion events necessitates the move towards a two-part MS2 system where T4 pol is delivered in trans. However, in the newly added data in Appendix Sup Fig 1L the authors show that there is no difference between whether a gRNA has an MS2 domain or not, and that simply the presence of T4 pol alone is enough to stimulate +2 bp insertions. This is contrary to the narrative of the study where it is implied that the MS2::MCP interaction is needed to bring T4 pol into the proximity of the staggered DSB to resolve it and impart its impact on indel spectra.

-In our study, to optimize the delivery approaches, we tested and compared three systems:

- A) MCP-tagged DNA polymerase, Cas9 and regular guide RNA (Page 6);
- B) MCP-tagged DNA polymerase, Cas9 and guide RNA with a MS2-modified scaffold (Page 8);
- C) A fusion of T4 DNA polymerase with Cas9 and regular guide RNA (Page 9)

For the delivery systems A and B, T4 DNA polymerase was delivered in trans. There is no significant difference between A and B (**Appendix Fig S1K**). These results indicate MCP-tag is dispensable for T4 DNA polymerase function when delivered in trans with Cas9 editing system in the cell lines tested. Therefore, all our in-vitro experiments unless otherwise noted in our study utilized MCP-tagged T4 DNA polymerase combined with regular guide RNA (system A).

We future clarified this point in the main text (Page 8).

Given Yoo et al clearly demonstrate the use of a DNA polymerase to counter large deletions and drive an increase in +1 bp insertions (and occasionally +2 bp insertions), to differentiate this study a more precise depiction of the underlying mechanisms is essential. The current manuscript provides minimal mechanistic insights, leaning towards speculation rather than empirical evidence. Data that delineate the mechanistic differences from the Yoo study would significantly bolster the novelty and scientific contribution of this paper.

-Many thanks for the insightful suggestion on underlying mechanism study, especially for the difference and similarity between T4 DNA polymerase (our study) and PolI/KF (Yoo et al).

1) New data: Local target sequence context highly influences the indel profile which was generated via Cas9-mediated editing (PMID:30405244, 27499295 and 24535568). Hence, we generated new data on *CLCN5* site which is the same guide RNA employed in Yoo et al's study:

- A) When delivered in trans, both T4/RB69 or PolI/KF enhanced 1-bp deletions, primary at the expense of relatively large deletions, at the *CLCN5* site (New panels in **appendix Fig S1 P**).
- B) In contrast, fusionT4-L-Cas9 and Cas9-L-T4 increased *relatively* large deletions (<25bp) at the PAM distal region at *CLCN5* site (New panels in **appendix Fig S1O-P**).

2) Mechanistic insights: T4 DNA polymerase possesses a higher 3'-to-5' exonuclease activity than PolI/KF (PMID: 18972387). Structure studies on the Cas9/guide RNA and targeted DNA complex reveal that the non-complementary 3' single-strand DNA strand are displaced upon formation of RNA-DNA R-loop (Jore, Lundgren et al., 2011 PMID: 21460843). Therefore, we speculated that the robust exonuclease domain within **T4 DNA polymerase, when fusion with Cas9**, might degrade the displaced 3' ssDNA at the PAM distal region, leading to PAM distal deletions (**New panel in appendix Fig S1Q**). We deactivated the T4 Pol exonuclease domain in substituting D219 to A219 (T4-D219A). Sequencing results demonstrated that the mutation of T4-D219A, but not T4-D214S which deactivate the DNA polymerase domain, eliminated the increased frequency of deletions at the PAM distal region at tdTomato-d151A site induced by T4-L-SpCas9 and Cas9-L-T4 relative to Cas9-only (**Appendix Fig S1R-S**). The improved deletions at the PAM distal region at *CLCN5* site were also abolished by introducing D219A mutation to T4-L-SpCas9 and Cas9-L-T4 (New panel in **appendix Fig S1T**). All these data support our

hypothesis that the robust 3'-to-5' exonuclease activity within T4 DNA polymerase partially contributes the differences observed between the fused T4 and Cas9 to fused PolI and Cas9.

Further characterization of the T4 DNA polymerase, Cas9, guide RNA and DNA crystal structure during DNA repair, although quite a challenge and beyond the scope of this paper, might elucidate this mechanism in more detail. More discussion is added in main text **Page 20**.

In summary, while the authors have made efforts to address previous concerns, the presentation and interpretation of the data require further clarity and distinction from prior studies to solidify the manuscript's contribution to the field.

-Again, we greatly appreciate this referee's comprehensive and in-depth suggestions. Hope this new revision with a considerable amount of new experiments and data will enhance the quality of our paper and will make a valuable contribution to the genome editing communities.

Dear Dr Long,

Thank you for submitting a revised version of your manuscript. Unfortunately, referee #3 did not respond to our request to re-review the manuscript. Your study has now been seen by the original referees #2, who finds that the previous concerns both by referee #2 and referee #3 have been addressed and now recommends publication of the manuscript. In addition, please find of a few mainly editorial points raised by our data editors that have to be addressed before I can extend formal acceptance of the manuscript:

1. MANUSCRIPT FORMAT: figures need to be removed from ms, figure legends should be listed below the References
2. FUNDING INFO: no info inserted in eJP, and should be: Departmental Start-Up Grant, NYU Langone Health, and Kids Connect Charitable Fund
3. Please add up to 5 keywords.
4. REFERENCE FORMAT: ok, alphabetical, but one ref has 20 authors + et al. instead of 10 authors + et al. (Stadtmauer EA, Fraietta JA, Davis MM, Cohen AD, Weber KL, Lancaster E, Mangan PA, Kulikovskaya I, Gupta M, Chen F, Tian L, Gonzalez VE, Xu J, Jung IY, Melenhorst JJ, Plesa G, Shea J, Matlawski T, Cervini A, Gaymon AL et al. (2020) CRISPR-engineered T cells in patients with refractory cancer. *Science* 367)
5. DATA AVAILABILITY SECTION: in, but should be renamed to "Data Availability"
6. COI: title needs renaming to "DISCLOSURE AND COMPETING INTERESTS STATEMENT"
7. AC/CRedit: section needs to be removed
8. FIGURE CALLOUTS: Tables EV1-EV9 should be called out as Appendix Table S1-S9
9. APPENDIX 1 FILE WITH ToC: Appendix file needs to be in PDF format; missing ToC with page numbers; nomenclature should be Appendix Figure S1-S6 and Appendix Table S1-S9 with the corresponding callouts
10. SOURCE DATA: missing SD for Fig. 3F, 4B, 5B, 6B and 6F; Please provide source data and also upload the filled in source data checklist. Source data files need to be saved in a scheme one figure/folder and then uploaded as .zip files. E.g. all the Source data files for figure 1 need to be saved in a single folder and this needs to be zipped and then uploaded as "SD figure 1.zip" file.
11. Synopsis:
Papers published in The EMBO Journal are accompanied online by a 'Synopsis' to enhance discoverability of the manuscript. It consists of A) a short (1-2 sentences) summary of the findings and their significance, B) 3-4 bullet points highlighting key results and C) a synopsis image that is 550x300-600 pixels large (width x height, jpeg or png format). You can either show a model or key data in the synopsis image. Please note that the image size is rather small and that text needs to be readable at the final size. Please send us this information together with the revised manuscript.
12. Figure Legends (main + EV): "1. Please note that information related to n is missing in the legends of figures 1e-g; 2c, f-g; 3c-d; 6c-e.
13. 2. Please note that the error bars are not defined in the legends of figures 1e-g; 2c, f, g; 3c-d; 6c-e."
14. Supplementary information" section should be removed from ms file.
We replaced Supplementary Information with Expanded View (EV) Figures and Tables that are collapsible/expandable online. EV Figures should be cited as 'Figure EV1, Figure EV2' etc. in the text and their respective legends should be included in the main text after the legends of regular figures. Further information on the format is available here:
<https://www.embopress.org/page/journal/14602075/authorguide#expandedview>.
15. Correct Section order: It should be: title page with complete author information, abstract, introduction, results, discussion, materials & methods, data availability section, acknowledgements, disclosure and competing interests statement, references, main figure legends, tables, expanded figure legends.

With best regards,

Cornelius Schneider

Cornelius Schneider, PhD
Editor | The EMBO Journal
c.schneider@embojournal.org

We realize that it is difficult to revise to a specific deadline. In the interest of protecting the conceptual advance provided by the work, we recommend a revision within 3 months (29th Aug 2024). Please discuss the revision progress ahead of this time with the editor if you require more time to complete the revisions. Use the link below to submit your revision:

Referee #2:

The previous issues have been sufficiently addressed.

All editorial and formatting issues were resolved by the authors.

Dear Dr. Long,

I am pleased to inform you that your manuscript has been accepted for publication in the EMBO Journal.

Yours sincerely,

Cornelius Schneider, PhD
Editor
The EMBO Journal
c.schneider@embojournal.org
